# Improving Offline RL by Blending Heuristics

**Sinong Geng**
Princeton University
Princeton, NJ

**Aldo Pacchiano**
Boston University
Broad Institute of
MIT and Harvard
Boston, MA

**Andrey Kolobov**
Microsoft Research
Redmond, WA

**Ching-An Cheng**
Microsoft Research
Redmond, WA

## Abstract

We propose **Heu**ristic **Bl**ending (HUBL), a simple performance-improving technique for a broad class of offline RL algorithms based on value bootstrapping. HUBL modifies the Bellman operators used in these algorithms, partially replacing the bootstrapped values with heuristic ones that are estimated with Monte-Carlo returns. For trajectories with higher returns, HUBL relies more on the heuristic values and less on bootstrapping; otherwise, it leans more heavily on bootstrapping. HUBL is very easy to combine with many existing offline RL implementations by relabeling the offline datasets with adjusted rewards and discount factors. We derive a theory that explains HUBL's effect on offline RL as reducing offline RL's complexity and thus increasing its finite-sample performance. Furthermore, we empirically demonstrate that HUBL consistently improves the policy quality of four state-of-the-art bootstrapping-based offline RL algorithms (ATAC, CQL, TD3+BC, and IQL), by 9% on average over 27 datasets of the D4RL and Meta-World benchmarks.

## 1 Introduction

Offline reinforcement learning (RL) aims to learn decision-making strategies from static logged datasets (Lange et al., 2012; Fujimoto et al., 2019). It has attracted increased interest in recent years, because the availability of large offline datasets are on the rise and online exploration required by alternative approaches such as online RL (Sutton and Barto, 2018) remains expensive and risky in many real-world applications, such as robotics and healthcare.

Among offline RL algorithms, we focus on model-free approaches using dynamic programming with value bootstrapping. These algorithms, including the commonly used CQL (Kumar et al., 2020), TD3+BC (Fujimoto and Gu, 2021), IQL (Kostrikov et al., 2022), and ATAC (Cheng et al., 2022), have demonstrated strong performance in offline RL benchmarks (Fu et al., 2020; Gulcehre et al., 2020). They follow the actor-critic scheme and adopt the principle of pessimism in the face of uncertainty to optimize an agent via a performance lower bound that penalizes taking unfamiliar actions. Despite their strengths, existing model-free offline RL methods also have a major weakness: they do not perform *consistently*. An algorithm that does well on one dataset may struggle on another, sometimes even underperforming behavior cloning (see Tarasov et al. (2022) and Appendix D.2). These performance fluctuations stand in the way of applying even the strongest offline RL approaches to practical problems.

In this work, we propose **Heu**ristic **Bl**ending (HUBL), an easy-to-implement technique to address offline RL's performance inconsistency. HUBL is an "aide" that operates *in combination with* a bootstrapping-based offline RL algorithm by using heuristic state value esti-

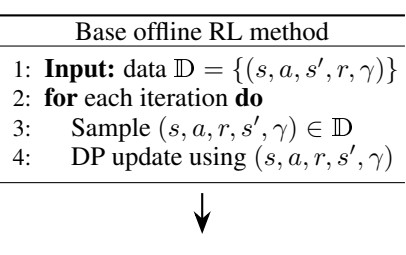

| Base offline RL method |
| --- |
| 1: **Input:** data $\mathbb{D} = \{(s, a, s', r, \gamma)\}$ |
| 2: **for** each iteration **do** |
| 3:     Sample $(s, a, r, s', \gamma) \in \mathbb{D}$ |
| 4:     DP update using $(s, a, r, s', \gamma)$ |

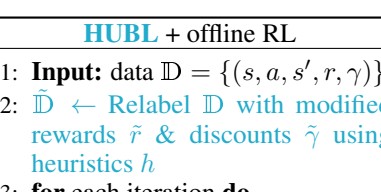

| **HUBL** + offline RL |
| --- |
| 1: **Input:** data $\mathbb{D} = \{(s, a, s', r, \gamma)\}$ |
| 2: $\tilde{\mathbb{D}} \leftarrow$ Relabel $\mathbb{D}$ with modified rewards $\tilde{r}$ & discounts $\tilde{\gamma}$ using heuristics $h$ |
| 3: **for** each iteration **do** |
| 4:     Sample $(s, a, \tilde{r}, s', \tilde{\gamma}) \in \tilde{\mathbb{D}}$ |
| 5:     DP update using $(s, a, \tilde{r}, s', \tilde{\gamma})$ |

Figure 1: HUBL and offline RL

mates[1] to modify the rewards and discounts in the dataset that the base offline RL algorithm consumes. Effectively, this modification blends heuristic values into dynamic programming to partially replace bootstrapping. Relying less on bootstrapping alleviates potential issues that bootstrapping causes and helps achieve more stable performance.

Combining HUBL with an offline RL method is very simple, as summarized in Figure 1, and amounts to running this method on a version of the original dataset with modified rewards $\tilde{r}$ and discounts $\tilde{\gamma}$: $\tilde{r} = r + \gamma\lambda h$ and $\tilde{\gamma} = \gamma(1 - \lambda)$, where $r$ is blended with a heuristic $h$, $\tilde{\gamma}$ is the reduced discount, and $\lambda \in [0, 1]$ is a blending factor representing the degree of trust towards the heuristic. We propose to set $h$ as the Monte-Carlo returns of the trajectories in the dataset for offline RL. Such heuristics are efficient and stable to compute, unlike bootstrapped $Q$-value estimates. The blending factor $\lambda$ in HUBL can be trajectory-dependent. Intuitively, we want $\lambda$ to be large (relying more on the heuristic) at trajectories where the behavior policy that collected the dataset performs well, and small (relying more on bootstrapped $Q$-values) otherwise. We provide three practical designs for $\lambda$; they use only one hyperparameter, which, empirically, does not need active tuning.

We analyze HUBL's performance both theoretically and empirically. Theoretically, we provide a finite-sample performance bound for a tabular offline RL with HUBL by framing it as solving a reshaped Markov decision process (MDP). To our knowledge, this is the first theoretical result for RL with heuristics in the *offline* setting. Our analysis shows that HUBL performs a bias-regret trade-off. On the one hand, solving the reshaped MDP with a smaller discount factor requires less bootstrapping and is relatively "easier", so the regret is smaller. On the other hand, HUBL induces bias due to reshaping the original MDP. Nonetheless, we demonstrate that the bias can be controlled by setting the $\lambda$ factor based on the above intuition, allowing HUBL to improve the performance of the base offline RL method.

Empirically, we run HUBL with the four aforementioned offline RL methods – CQL, TD3+BC, IQL, and ATAC – and show that enhancing these SoTA algorithms with HUBL can improve their performance by $9\%$ on average across 27 datasets of D4RL (Fu et al., 2020) and Meta-World (Yu et al., 2020). Notably, in some datasets where the base offline RL method shows inconsistent performance, HUBL can achieve more than 50% relative performance improvement.

## 2  RELATED WORK

**Bootstrapping-based offline RL**  A fundamental challenge of bootstrapping-based offline RL is the *deadly triad* (Sutton and Barto, 2018): a negative interference between *1)* off-policy learning from data with limited support, *2)* value bootstrapping, and *3)* the function approximator. Modern offline RL algorithms such as CQL (Kumar et al., 2020), TD3+BC (Fujimoto and Gu, 2021), IQL (Kostrikov et al., 2022), ATAC (Cheng et al., 2022), PEVI (Jin et al., 2021), MOReL (Kidambi et al., 2020), and VI-LCB (Rashidinejad et al., 2021) employ pessimism to discourage the agent from taking actions unsupported by the data, which has proved to be an effective strategy to address the issue of limited support. However, they still suffer from a combination of errors in bootstrapping and function approximation. HUBL aims to address this with stable-to-compute Monte-Carlo return heuristics and thus is a *complementary* technique to pessimism.

**RL by blending multi-step returns**  The idea of blending multi-step Monte-Carlo returns into the bootstrapping operator to reduce the degree of bootstrapping and thereby increase its performance has a long history in RL. This technique has been widely used in temporal difference methods (Sutton and Barto, 2018), where the blending is achieved by modifying gradient updates (Seijen and Sutton, 2014; Sutton et al., 2016; Jiang et al., 2021) and reweighting observations (Imani et al., 2018). It has also been applied to improve Q function estimation (Wright et al., 2013), online exploration (Ostrovski et al., 2017; Bellemare et al., 2016) and the sensitivity to model misspecification (Zanette et al., 2021), and is especially effective for sparse-reward problems Wilcox et al. (2022). In contrast to most of the aforementioned works, which focus on blending Monte-Carlo returns as part of an RL algorithm's *online* operation, HUBL is designed for the *offline* setting and acts as a simple data relabeling step. Recently, Wilcox et al. (2022) have also proposed the idea of data relabeling, but their design takes a max of multi-step returns and bootstrapped values and as a result tends to overestimate $Q$-functions. We observed this to be detrimental when data has a limited support.

---

[1] In this work, a heuristic is a mapping from states to $\mathbb{R}$ (Mausam and Kolobov, 2012). See Section 3.2.

**RL with heuristics** More generally, HUBL relates to the framework of blending heuristics (which might be estimating quantities other than a policy's value) into bootstrapping (Cheng et al., 2021; Hoeller et al., 2020; Bejjani et al., 2018; Zhong et al., 2013). However, the existing results focus only on the online case and do not conclusively show whether blending heuristics is valid in the offline case. For instance, the theoretical analysis in Cheng et al. (2021) breaks when applied to the offline setting as we will demonstrate in Section 5. The major difference is that online approaches rely heavily on collecting new data with the learned policy, which is impossible in the offline case. In this paper, we employ a novel analysis technique to demonstrate that blending heuristics is effective even in offline RL. To the best of our knowledge, ours is the first work to extend heuristic blending to the offline setting with both rigorous theoretical analysis and empirical results demonstrating performance improvement. One key insight, inspired by transition-dependent discount factor from White (2017), is the adoption of a *trajectory-dependent* $\lambda$ blending factor, which is both a performance-improving design as well as a novel analysis technicality.

**Discount regularization** HUBL modifies the reward with a heuristic *and* reduces the discount. Discount regularization, on the other hand, is a complexity reduction technique that reduces *only* the discount. The idea of simplifying decision-making problems by reducing discount factors can be traced back to Blackwell optimality in the known MDP setting (Blackwell, 1962). Most existing results on discount regularization (Petrik and Scherrer, 2008; Jiang et al., 2015) study the MDP setting or the online RL setting (Van Seijen et al., 2019). Recently, Hu et al. (2022) has shown that discount regularization can also reduce the complexity of offline RL and serve as an extra source of pessimism. However, as we will show, simply reducing the discount without the compensation of blending heuristics in the offline setting can be excessively pessimistic and introduce large bias that hurts performance. In Section 6.1, we rigorously analyze this bias and empirically demonstrate the advantages of HUBL over discount regularization.

## 3 BACKGROUND

In this section, we define the problem setup of offline RL (Section 3.1), review dynamic programming with value bootstrapping, and briefly survey methods using heuristics (Section 3.2).

### 3.1 OFFLINE RL AND NOTATION

We consider offline RL in a Markov decision process (MDP) $\mathcal{M} = (\mathcal{S}, \mathcal{A}, \mathcal{P}, r, \gamma)$, where $\mathcal{S}$ denotes the state space, $\mathcal{A}$ denotes the action space, $\mathcal{P}$ denotes the transition function, $r : \mathcal{S} \times \mathcal{A} \to [0, 1]$ denotes the reward function, and $\gamma \in [0, 1)$ denotes the discount factor. A decision-making *policy* $\pi$ is a mapping from $\mathcal{S}$ to $\mathcal{A}$, and its *value function* is defined as $V^\pi(s) = \mathbb{E}[\sum_{t=0}^{\infty} \gamma^t r(s_t, a_t)|s_0 = s, a_t \sim \pi(\cdot|s_t)]$. In addition, we define $Q^\pi(s, a) := r(s, a) + \gamma \mathbb{E}_{s' \sim \mathcal{P}|s,a}[V^\pi(s')]$ as its state-action value function (i.e., Q function). We use $V^*$ to denote the value function of the optimal policy $\pi^*$, which is our performance target.

In addition, we introduce several definitions of distributions. First, we define the average state distribution of a policy $\pi$ starting from an initial state $s_0$ as $d^\pi(s, a; s_0) := (1 - \gamma) \sum_{t=0}^{\infty} \gamma^t d_t^\pi(s, a; s_0)$, where $d_t^\pi(s, a; s_0)$ is the state-action distribution at time $t$ generated by running policy $\pi$ from an initial state $s_0$. We assume that the MDP starts with a fixed initial state distribution $d_0$. With slight abuse of notation, we define the average state distribution starting from $d_0$ as $d^\pi(s, a) := d^\pi(s, a; d_0)$, and $d^\pi(s, a, s') := d^\pi(s, a)\mathcal{P}(s'|s, a)$.

The objective of offline RL is to learn a well-performing policy $\hat{\pi}$ while using a pre-collected offline dataset $\mathbb{D}$. The agent has no knowledge of the MDP $\mathcal{M}$ except information contained in $\mathbb{D}$, and it cannot perform online interactions with environment to collect more data. We assume $\mathbb{D} := \{\tau\}$ contains multiple trajectories collected by a *behavior policy*, where each $\tau = \{(s_t, a_t, r_t)\}_{t=1}^{T_\tau}$ denotes a trajectory with length $T_\tau$. Suppose these trajectories contain $N$ transition tuples in total. With abuse of notation, we also write $\mathbb{D} := \{(s, a, s', r, \gamma)\}$, where states $s$ and action $a$ follow a distribution $\mu(s, a)$ induced by the behavior policy, $s'$ is the state after each transition, $r$ is the reward at $s, a$, and $\gamma$ is the discount factor of the MDP. Note that the value of discount $\gamma$ is the same in each tuple. We use $\Omega$ to denote the support of $\mu(s, a)$. *We do not make the full support assumption, in the sense that the dataset $\mathbb{D}$ may not contain a tuple for every $s, a, s'$ transition in the MDP.*

## 3.2 Dynamic Programming with Bootstrapping and Heuristics

**Bootstrapping** Many offline RL methods leverage dynamic programming with value bootstrapping. Given a policy $\pi$, we recall $Q^\pi$ and $V^\pi$ satisfy the Bellman equation:

$$Q^\pi(s, a) = r(s, a) + \overbrace{\gamma \mathbb{E}_{s' \sim \mathcal{P}(\cdot|s,a)}[V^\pi(s')]}^{\text{(bootstrapping)}}. \tag{1}$$

These bootstrapping-based methods compute $Q^\pi(s, a)$ using an approximated version of (1): given sampled tuples $(s, a, s', \gamma, r)$, such methods minimize the difference between the two sides of (1) with both $Q^\pi$ and $V^\pi$ replaced with function approximators [2]. With limited offline data, learning the function approximator using bootstrapping can be challenging and yields inconsistent performance across different datasets (Dulac-Arnold et al., 2021; Sutton and Barto, 2018; Kumar et al., 2019), as we will also later see in the experiment section (Section 6).

**Heuristics** A *heuristic* is a value function $h : \mathcal{S} \rightarrow \mathbb{R}$ calculated using domain knowledge, Monte-Carlo averages, or pre-training. Heuristics are widely used in online RL, planning, and control to improve the performance of decision-making (Kolobov et al., 2010; Zhong et al., 2013; Bejjani et al., 2018; Hoeller et al., 2020; Cheng et al., 2021). In this paper, we focus on the offline setting, and consider heuristics $h$ that approximate the value function of the behavior policy. They can be estimated from $\mathbb{D}$ via Monte-Carlo methods.

## 4 Heuristic Blending (HUBL)

In this section we describe our main contribution — **Heu**ristic **Bl**ending (HUBL), an algorithm that works in combination with bootstrapping-based offline RL methods and improves their performance.

### 4.1 Motivation

HUBL uses a heuristic computed as the Monte-Carlo return of the behavior policy in the training dataset $\mathbb{D}$ to improve offline RL. It reduces an offline RL algorithm's bootstrapping at trajectories where the behavior policy performs well, i.e., where the value of the behavior policy (i.e. the heuristic value) is high. With less amount of bootstrapping, it mitigates bootstrapping-induced issues on convergence stability and performance. In addition, since the extent of

---

**Algorithm 1** HUBL + Offline RL

1: **Input:** Dataset $\mathbb{D} = \{(s, a, s', r, \gamma)\}$
2: Compute $h_t$ for each trajectory in $\mathbb{D}$
3: Compute $\lambda_t$ for each trajectory in $\mathbb{D}$
4: Relabel $r$ & $\gamma$ by $h_t$ and $\lambda_t$ as $\tilde{r}$ and $\tilde{\gamma}$ and create $\tilde{\mathbb{D}} = \{(s, a, s', \tilde{r}, \tilde{\gamma})\}$
5: $\hat{\pi} \leftarrow$ Offline RL on $\tilde{\mathbb{D}}$

---

blending between the heuristic and bootstrapping is trajectory-dependent, HUBL introduces only limited performance bias to the base algorithm, and therefore can improve its performance overall.

### 4.2 Algorithm

As summarized in Algorithm 1, HUBL is easy to implement: first, relabel a base offline RL algorithm's training dataset $\mathbb{D} = \{(s, a, s', r, \gamma)\}$ with modified rewards $\tilde{r}$ and discount factors $\tilde{\gamma}$, creating a new dataset $\tilde{\mathbb{D}} := \{(s, a, s', \tilde{r}, \tilde{\gamma})\}$; next, run the base algorithm on $\tilde{\mathbb{D}}$.

The data relabeling is done in three steps:

**Step 0:** As a preparation step, we convert the data tuples in $\mathbb{D}$ back to trajectories like $\tau = \{(s_t, a_t, r_t)\}_{t=1}^{T_\tau}$. For the next two steps, we work on data trajectories instead of data tuples to compute heuristics and blending factors.

**Step 1: Computing heuristic** $h_t$ We compute heuristics by Monte-Carlo returns. For each $\tau = \{(s_t, a_t, r_t)\}_{t=1}^{T_\tau} \in \mathbb{D}$, we calculate the heuristics as [3] $h_t = \sum_{k=t}^{T_\tau} \gamma^{k-t} r_k$ and update the data trajectory as $\tau \leftarrow \{(s_t, a_t, r_t, h_t)\}_{t=1}^{T_\tau}$.

---

[2] Q-learning-based offline RL in Kostrikov et al. (2022) uses a dynamic programming equation similar to (1) but with $V^\pi(s) = \arg\max_{a \in \mathcal{A}} Q^\pi(s, a)$.

[3] In our implementation, we also train a value function approximator to bootstrap at the *end* of a trajectory if the trajectory ends due to timeout as opposed to reaching the end of the problem horizon or a terminal state.

**Step 2: Computing blending factor** $\lambda_t$  We append a scalar $\lambda_t = \lambda(\tau) \in [0, 1]$ at each time point $t$ of each trajectory as the blending factor, leading to $\tau \leftarrow \{(s_t, a_t, r_t, h_t, \lambda_t)\}_{t=1}^{T_\tau}$. $\lambda_t$ indicates the confidence in the heuristics on the trajectory. Intuitively, $\lambda_t$ decides the contribution of heuristics over bootstrapped values in dynamic programming to update the $Q$-function. We desire $\lambda_t$ to be closer to 1 when the heuristic value $h_t$ is higher (i.e., at states where the heuristic is closer to the optimal $Q$-value) to make offline RL rely more on the heuristic, and $\lambda_t$ closer to zero when heuristic is lower to make offline RL to use more bootstrapping. We experiment with three different designs of $\lambda(\tau)$:

- *Constant*: As a baseline, we consider $\lambda(\tau) = \alpha \in [0, 1]$ for all $s$. We show that, despite forcing the same heuristic weight for every state, this formulation already provides performance improvements.

- *Sigmoid*: As an alternative, we use the sigmoid function to construct a trajectory-dependent blending function $\lambda(\tau) = \alpha\sigma(\sum_{t=1}^{T_\tau} h(s_t)/T_\tau)$, where $\alpha \in [0, 1]$ is a tunable constant and $\sigma$ is the sigmoid function. Thus, $\lambda(\tau)$ varies with the performance of the behavior policy over data.

- *Rank*: Similar to the Sigmoid labeling function, we provide a rank labeling function $\lambda(\tau) = \alpha \sum_{\tau' \in \mathbb{D}} \mathbb{1}_{\bar{h}(\tau') \leq \bar{h}(\tau)}/n$ where $n$ is the number of trajectories in $\mathbb{D}$, and $\bar{h}(\tau) = \frac{1}{T}\sum_{h_t \in \tau} h_t$.

**Step 3: Relabeling** $r$ **and** $\gamma$  Finally, we relabel the reward as $\tilde{r}$ and the discount factor as $\tilde{\gamma}$ in each tuple of $\mathbb{D}$. To this end, we first convert the updated data trajectories $\{\{(s_t, a_t, r_t, h_t, \lambda_t)\}_{t=1}^{T_\tau}\}$ back into data tuples $\{(s, a, s', r, \gamma, h', \lambda')\}$, where $h'$ and $\lambda'$ denote the next-step heuristic and blending factor. Then, for each data tuple, we compute

$$\tilde{r} = r + \gamma\lambda'h' \qquad \text{and} \qquad \tilde{\gamma} = \gamma(1 - \lambda'), \qquad (2)$$

to form a new dataset $\tilde{\mathbb{D}} := \{(s, a, s', \tilde{r}, \tilde{\gamma})\}$. Intuitively, one can interpret $\tilde{r}$ as injecting a heuristic-dependent quantity $\gamma\lambda'h'$ into the original reward, and $\tilde{\gamma}$ as reducing bootstrapping by shrinking the original discount factor by a factor of $1 - \lambda'$. We formally justify the design of $\tilde{r}$ and $\tilde{\gamma}$ in Section 5.

## 5 UNDERSTANDING HUBL

In this section, we take a deeper look into how HUBL works. At a high level, our theoretical analysis explains that the modification made by HUBL introduces a trade-off between bias and regret (which is similar to the variance of policy learning) into the base offline RL algorithm. While an important part of our analysis is for tabular settings (Section 5.3), we believe it still provides valuable intuitions as to why reshaping of rewards and discounts per (2) gives a performance boost to state-of-the-art offline RL algorithms in the continuous-state-space benchmarks in Section 6.

### 5.1 HUBL AS MDP RESHAPING

We analyze HUBL by viewing it as solving a reshaped MDP $\tilde{\mathcal{M}} := (\mathcal{S}, \mathcal{A}, \mathcal{P}, \tilde{r}, \tilde{\gamma})$ constructed by blending heuristics into the original MDP. To this end, we make a simplification by assuming that both the heuristic and blending factor are functions of states. Specifically, let $\Omega$ denote the support of the data distribution; we suppose that some functions $h(\cdot) : \Omega \to \mathbb{R}$ and $\lambda(\cdot) : \Omega \to [0, 1]$ are given to HUBL. For analysis, we extend them to out-of-$\Omega$ states as below:

$$h(s) = \begin{cases} h(s) & \text{for } s \in \Omega \\ 0 & \text{otherwise} \end{cases} \quad \text{and} \quad \lambda(s, s') = \begin{cases} \lambda(s') & \text{for } s, s' \in \Omega \\ 0 & \text{otherwise} \end{cases} \qquad (3)$$

We note that this extension is only for the purpose of analysis, since HUBL never uses the values $h(\cdot)$ and $\lambda(\cdot)$ outside $\Omega$; the given $h$ on out-of-$\Omega$ can have any value and the following theorems still hold.

We define the reshaped MDP $\tilde{\mathcal{M}}$ with redefined reward function and discount factor as $\tilde{r}(s, a) := r(s, a) + \gamma\mathbb{E}_{s' \sim \mathcal{P}(\cdot|s,a)}[\lambda(s, s')h(s')]$, and $\tilde{\gamma}(s, s') := \gamma(1 - \lambda(s, s'))$ respectively. Note that we blend the original reward function with the expected heuristic and blending factor while adjusting the original discount factor correspondingly. The extent of blending is determined by the function $\lambda(\cdot)$. Notice that this reshaped MDP has a transition-dependent[4] discount factor $\tilde{\gamma}(s, s')$ which is compatible with generic unifying task specification of (White, 2017). This novel definition of

---

[4]A special case of trajectory-dependent discounts.

transition-dependent discount factor is a key analysis technique to show that blending heuristics is effective in the *offline* setting.

**Reshaped Dynamic Programming**  When solving this reshaped MDP by dynamic programming, the Bellman equation changes from (1) accordingly into

$$\tilde{Q}^\pi(s,a) = \tilde{r}(s,a) + \mathbb{E}_{s' \sim \mathcal{P}(\cdot|s,a)}[\tilde{\gamma}(s,s')\tilde{V}^\pi(s')]$$

$$= r(s,a) + \gamma \overbrace{\mathbb{E}_{s' \sim \mathcal{P}(\cdot|s,a)}[(1-\lambda(s,s'))\tilde{V}^\pi(s')]}^{\text{bootstrapping}} + \gamma \overbrace{\mathbb{E}_{s' \sim \mathcal{P}(\cdot|s,a)}[\lambda(s,s')h(s')]}^{\text{heuristic}}. \qquad (4)$$

Here $\tilde{Q}^\pi$ denotes the $Q$-function of policy $\pi$ in $\tilde{\mathcal{M}}$, and $\tilde{V}^\pi$ denotes $\pi$'s value function. Compared to the original Bellman equation (1), it can be seen that $\lambda(\cdot)$ blends the heuristic with bootstrapping: the bigger $\lambda$'s values, the more bootstrapping is replaced by the heuristic. The effect of solving HUBL's reshaped MDP $\tilde{\mathcal{M}}$ is twofold. On the one hand, $\tilde{\mathcal{M}}$ is different from the original MDP $\mathcal{M}$: the optimal policy for $\tilde{\mathcal{M}}$ may not be optimal for $\mathcal{M}$, so solving for $\tilde{\mathcal{M}}$ could potentially lead to performance bias. One the other hand, $\tilde{\mathcal{M}}$ has a smaller discount factor and thus is easier to solve than $\mathcal{M}$, as the agent needs to plan for a smaller horizon. Therefore, we can think of applying HUBL to offline RL problems as performing a bias-variance trade-off, which reduces the learning variance due to bootstrapping at the cost of the bias due to using a suboptimal heuristic. We will explain this more concretely next.

## 5.2 BIAS-REGRET DECOMPOSITION

The insight that HUBL reshapes the Bellman equation that the offline RL algorithm uses allows us to characterize HUBL's effects on policy learning. Namely, we use the modified Bellman equation from (4) to decompose the performance of the learned policy into bias and regret terms.

**Theorem 1.** *For any $h : \Omega \to \mathbb{R}$, $\lambda : \Omega \to [0,1]$, and policy $\pi$, with $V^*$ as the value function of the optimal policy, it holds that $V^*(d_0) - V^\pi(d_0) = \text{Bias}(\pi, h, \lambda) + \text{Regret}(\pi, h, \lambda)$, where*

$$\text{Bias}(\pi, h, \lambda) := \frac{\gamma}{1-\gamma}\mathbb{E}_{(s,a,s') \sim d^{\pi^*}}[\lambda(s')(\tilde{V}^{\pi^*}(s') - h(s'))|s, s' \in \Omega]$$

$$\text{Regret}(\pi, h, \lambda) := \tilde{V}^{\pi^*}(d_0) - \tilde{V}^\pi(d_0) + \frac{\gamma}{1-\gamma}\mathbb{E}_{(s,a,s') \sim d^\pi}[\lambda(s')(h(s') - \tilde{V}^\pi(s'))|s, s' \in \Omega].$$

The performance of $\pi$ depends on both bias and regret. The bias term describes the discrepancy caused by solving the reshaped MDP with $\lambda(\cdot)$. When $\lambda(s') = 0$, the bias becomes zero. The regret term describes the performance of the learned policy in the reshaped MDP. Intuitively, at states whose successors have bigger $\lambda(s')$ values, the reshaped MDP has a smaller discount factor and thus is easier to solve, which leads to smaller regret (i.e., smaller values for $\tilde{V}^{\pi^*}(d_0) - \tilde{V}^{\hat{\pi}}(d_0)$). Therefore, solving a HUBL-reshaped MDP induces a bias term but may generate a smaller regret.

**Remark**  The critical difference – and novelty – of Theorem 1 compared to existing theoretical results for RL with heuristics in the offline setting is that both the bias and regret in Theorem 1 depend only on states *in the data distribution support* $\Omega$. This is crucial, because in the offline setting we have no access to observations beyond $\Omega$. In contrast, if in the preceding analysis we replace Theorem 1 by, for example, Lemma A.1 from Cheng et al. (2021) with a constant $\lambda$, we will get a performance decomposition $V^*(d_0) - V^\pi(d_0) = (V^*(d_0) - \tilde{V}^*(d_0)) + \frac{\gamma\lambda}{1-\gamma}\mathbb{E}_{s,a \sim d^\pi}\mathbb{E}_{s'|s,a}[h(s') - \tilde{V}^*(s')]$ $+ (1-\lambda)(\tilde{V}^*(d_0) - \tilde{V}^\pi(d_0)) + \frac{\lambda}{1-\gamma}(\tilde{V}^*(d^\pi) - \tilde{V}^\pi(d^\pi))$. The decomposition, however, suggests that the out-of-$\Omega$ values of $\lambda(s)$ or $h(s)$ are important to the performance of HUBL.

## 5.3 FINITE-SAMPLE ANALYSIS FOR BIAS AND REGRET

The finite-sample analysis of policy learning that we provide next illustrates more concretely how HUBL trades off bias and regret. Our analysis uses offline value iteration with lower confidence bound (VI-LCB) (Rashidinejad et al., 2021) as the base offline RL method. Following its original convention, we make some technical simplifications to make the presentation cleaner. Specifically, we consider a tabular setting with $\lambda(s) = \alpha \in [0,1]$ for $s \in \Omega$ as a constant value, which can be interpreted as the quality of the behavior policy averaged across states. Note that although $\lambda(s) = \alpha$

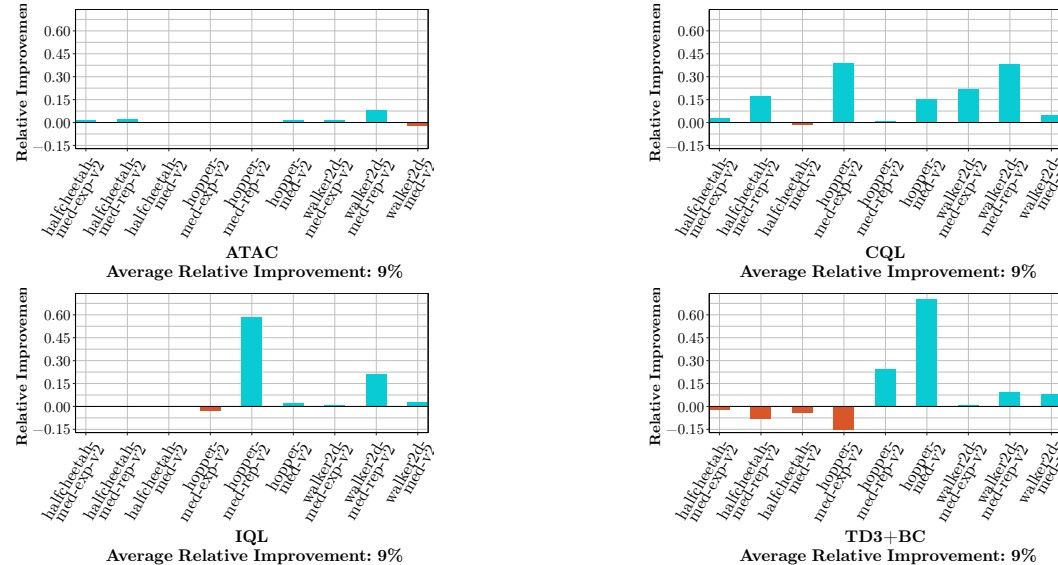

Figure 2: Relative improvement of HUBL with rank blending on 9 D4RL datasets.

is a constant on $\Omega$, the reshaped MDP is still defined by $\lambda(s, s')$ in (3) (i.e., $\lambda(s, s') = \alpha$ if $s, s' \in \Omega$ and zero otherwise). The details of VI-LCB with HUBL are in Appendix C due to space limits.

Theorem 2 summarizes our finite-sample results. The proof of Theorem 2 can be found in Appendix B.

**Theorem 2.** *Under the setup described above, assume that the heuristic $h(\cdot)$ satisfies $h(s) = V^\mu(s)$ for any $s \in \Omega$. Then the bias and the regret in Theorem 1 are bounded by*

$$\text{Bias}(\hat{\pi}, \lambda) \leq \frac{\gamma\lambda}{1-\gamma} \mathbb{E}_{(s,a,s') \sim d^{\pi^*}}[V^*(s') - V^\mu(s')|s, s' \in \Omega],$$

$$\mathbb{E}_{\mathbb{D}}[\text{Regret}(\hat{\pi}, \lambda)] \lesssim \min \left( V_{max}, \sqrt{\frac{V_{max}^2(1-\gamma)|\mathcal{S}|}{N(1-\gamma(1-\lambda))^4}} \left( \sqrt{\max_{s,a} \frac{d^{\pi^*}(s,a)}{\mu(s,a)}} + \frac{\gamma\lambda}{1-\gamma} \sqrt{\max_{(s,a) \in \Omega} \frac{1}{\mu(s,a)}} \right) \right),$$

*where $V_{max}$ denotes a constant upper bound for the value function.*

For the bias bound, the assumption $h(s) = V^\mu(s)$ for any $s \in \Omega$ is made for the ease of presentation. If it does not hold, an additive error term can be introduced in the bias bound to capture that. For the regret bound, $\max_{s,a} \frac{d^{\pi^*}(s,a;d_0)}{\mu(s,a)}$ can be infinite, whereby the best regret bound is just $V_{max}$. But when it is bounded as assumed by existing works (Rashidinejad et al., 2021), our results demonstrate how $N$, $\gamma$ and $\lambda$ affect the regret bound.

The implications of Theorem 2 are threefold. First of all, it provides a finite-sample performance guarantee for HUBL with VI-LCB under the tabular setting. Compared with the performance bound of the original VI-LCB, $\min \left( V_{max}, \sqrt{\frac{V_{max}^2|\mathcal{S}|}{(1-\gamma)^3 N} \max_{s,a} \frac{d^{\pi^*}(s,a)}{\mu(s,a)}} \right)$, HUBL shrinks the discount factor by $1 - \lambda$ and thus potentially improves the performance while inducing bias. Second, Theorem 2 hints at the source of HUBL's bias and regret of HUBL. The bias is related to the the performance of the behavior (data-collection) policy, characterized by $V^*(s) - V^\mu(s)$. In the extreme case of data being collected by an expert policy, the bias induced by HUBL is 0. The regret is affected by $\frac{d^{\pi^*}(s,a)}{\mu(s,a)}$, which describes the deviation of the optimal policy from the data distribution. Finally, Theorem 2 also provides guidance on how to construct a blending factor function $\lambda(\cdot)$. To reduce bias, $\lambda(s)$ should be small at states where $V^*(s) - V^\mu(s)$ is small. To reduce regret, $\lambda(\cdot)$ should be generally large but small at states where the learned policy is likely to deviate from the behavior policy. Therefore, an ideal $\lambda(\cdot)$ should be large when the behavior policy is close to optimal but small when it deviates from the optimal policy. This is consistent with our design principle in Section 4.2.

**Extension to Model-based Offline RL** Conceptually, we can implement HUBL with model-based offline RL methods by directly relabeling the data, learning the heuristic-modified reward model,

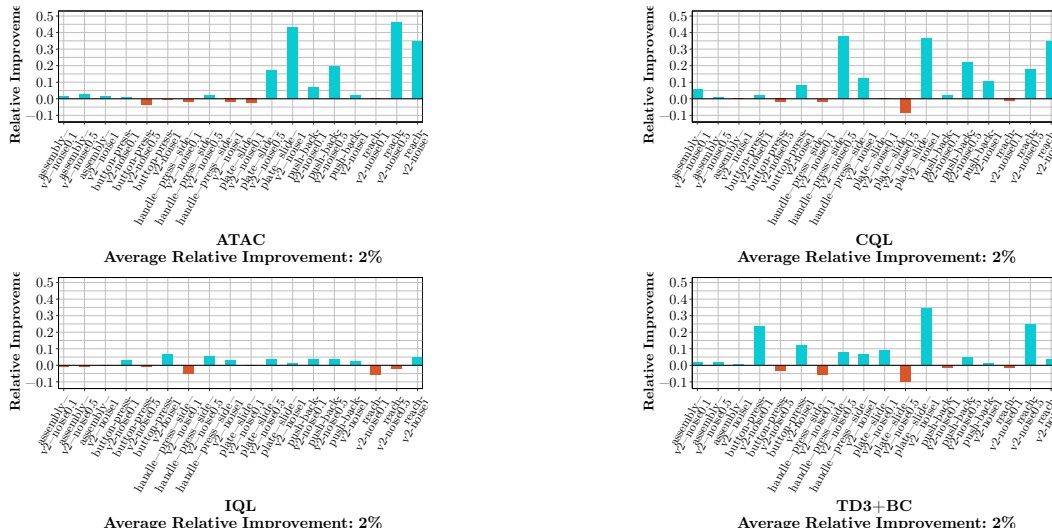

Figure 3: Relative improvement of HUBL with rank labeling on MW datasets.

and then doing model-based planning with a smaller discount, following (2). Theorem 1 still applies to model-based algorithms. For Theorem 2, we expect that a similar analysis can be applied to model-based algorithms like MOReL. We defer this direction to future work and focus on improving the stability of a wide range of model-free offline RL methods.

## 6 EXPERIMENTS

We study 27 benchmark datasets in D4RL and Meta-World. We show that HUBL improves the performance of existing offline RL methods by $9\%$ with easy modifications and simple hyperparameter tuning. Remarkably, in some datasets where the base offline RL performs inconsistently or poorly, HUBL can achieve more than 50% performance improvement. When the dataset is simple and the base offline RL is already near-optimal, HUBL does not significantly harm the performance.

**HUBL variants and base offline RL methods** We implement HUBL with four state-of-the-art offline RL algorithms as base methods: CQL (Kumar et al., 2020), TD3+BC (Fujimoto and Gu, 2021), IQL (Kostrikov et al., 2022), and ATAC (Cheng et al., 2022). For each base method, we compare the performance of its original version to its performance with HUBL running three different blending strategies discussed in Section 4.2: constant, sigmoid and rank. Thus, we experiment with 16 different methods in total. The implementation details of the base methods are in Appendix D.1.

**Metrics** We use relative normalized score improvement, abbreviated as *relative improvement*, as a measure of HUBL's performance improvement. Specifically, for a given task and base method, we first compute the normalized score $r_{base}$ achieved by the base method, and then the normalized score $r_{HUBL}$ of HUBL. The relative normalized score improvement is defined as $\frac{r_{HUBL}-r_{base}}{|r_{base}|}$. We report the relative improvement of HUBL averaged over three seeds $\{0, 1, 10\}$ in this section, with *standard deviations, base method performance, behavior cloning performance and absolute normalized scores* provided in Appendix D.2, D.5, and D.7.

**Hyperparameter Tuning** For each dataset, the hyperparameters of the base methods are tuned over six different configurations suggested by the original papers. HUBL has one extra hyperparameter, $\alpha$, which is *fixed* for all the datasets but different for each base method. Specifically, $\alpha$ is selected from $\{0.001, 0.01, 0.1\}$ according to the relative improvement averaged over all the datasets. In practice, we notice that a single choice of $\alpha$ around $0.1$ is sufficient for good performance across most base offline RL methods and datasets as demonstrated by the sensitivity analysis in Appendix D.3.

## 6.1 PERFORMANCE IMPROVEMENT OF HUBL

We study 9 benchmark datasets in D4RL and 18 datasets for tasks in Meta-World, collected following the procedure detailed in Appendix D.4. The relative improvement of HUBL with the rank labeling method is reported in Figure 2 and Figure 3. First, despite being simple and needing little hyperparameter tuning, HUBL improves the performance of base methods in most settings and only slightly hurts in some expert datasets where base offline RL methods are already performing very well with little space for improvement. Second, there are cases where HUBL achieve very significant relative improvement—more than 50%. Such big improvement happens in the datasets where the base method shows inconsistent performance and underperforms other offline RL algorithms. HUBL solves this inconsistent performance issue by simply relabeling the data. Further, HUBL improves performance even on data with few expert trajectories like hopper-med-rep-v2, walker2d-med-rep-v2, and hopper-med-v2, because HUBL conducts more bootstrapping and relies less on the heuristic on suboptimal trajectories (see Section 4.2).

## 6.2 ABLATION STUDIES

**Discount Relabeling** HUBL relabels *both* the reward *and* the discount factor, per (2). In contrast, existing methods like Hu et al. (2022) suggest that a lower discount factor *alone*, without blending heuristics into rewards, can in general improve offline RL. To assess the need for modifying both the discount factor and rewards like (2), we consider ablation methods which shrink *only* the discount factor as $\tilde{\gamma}$ *without* $\tilde{r}$. The achieved average relative improvement of these ablations is reported and compared with that of HUBL in Table 2. HUBL consistently outperforms these ablations, which justifies HUBL's coordinated modifications in both the reward and the discount factor. The advantage of HUBL is also consistent with what our theoretical analysis predicts. The considered ablations do not modify the rewards and thus are equivalent to solving $\tilde{Q}^\pi(s,a) = r(s,a) + \gamma\mathbb{E}_{s'\sim\mathcal{P}(\cdot|s,a)}[(1-\lambda(s'))\tilde{V}^\pi(s')]$. Comparing it with (1), the solution will be consistently smaller than the true $Q$-function, inducing a pessimistic bias. Crucially, this bias is much more challenging to tackle than the bias induced by HUBL, because the former is inevitable even when the data is from an expert policy.

|  | Sigmoid | Rank | Constant |
|---|---|---|---|
| Wins | 30 | 46 | 32 |

Table 1: Blending strategy comparison

| **ATAC** | Sigmoid | Rank | Constant |
|---|---|---|---|
| Ablations without $\tilde{r}$ | -0.01 | -0.02 | -0.02 |
| HUBL | 0.01 | 0.02 | 0.01 |
| **CQL** | Sigmoid | Rank | Constant |
| Ablations without $\tilde{r}$ | 0.04 | 0.04 | 0.04 |
| HUBL | 0.11 | 0.16 | 0.13 |
| **IQL** | Sigmoid | Rank | Constant |
| Ablations without $\tilde{r}$ | -0.04 | -0.04 | -0.07 |
| HUBL | 0.09 | 0.09 | 0.06 |
| **TD3+BC** | Sigmoid | Rank | Constant |
| Ablations without $\tilde{r}$ | -0.01 | -0.03 | -0.04 |
| HUBL | 0.06 | 0.09 | 0.1 |

Table 2: Average relative improvement of ablations without $\tilde{r}$.

**Comparison of Blending Strategies** We present results for HUBL with rank blending, while the results of other blending strategies (sigmoid and constant) are provided in Appendix D.2 and D.5. We notice that the rank blending outperforms the other two. With 27 datasets and 4 base methods, we have 108 cases. In Table 1 we report the number of cases where a given blending strategy provides the best performance among the three. We can see that rank is favored on average.

We also experiment HUBL in other different settings, including on datasets collected by expert policies (Appendix D.6.1), with online RL methods (Appendix D.6.2), and in stochastic environments (Appendix D.6.3). HUBL shows consistent performance improvements in these settings.

## 7 CONCLUSION

In this work, we propose HUBL, a method for improving the performance of offline RL methods by blending heuristics with bootstrapping. HUBL is easy to implement and is generally applicable to various of offline RL methods. Empirically, we demonstrate the performance improvement of HUBL on 27 datasets in D4RL and Meta-World. We also provide a theoretic finite-sample performance bound for HUBL, which sheds lights on HUBL's bias-regret trade-off and blending factor designs.

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

## A    Extended Results of Theorem 1

Below we prove the statement which is a restatement of theorem 1.

**Theorem 3.** *For any* $\lambda : \Omega \to [0,1]$ *and any* $h : \Omega \to \mathbb{R}$*, it holds*

$$V^*(s_0) - V^{\hat{\pi}}(s_0) = \left( \tilde{V}^{\pi^*}(s_0) - \tilde{V}^{\hat{\pi}}(s_0) \right) \tag{5}$$
$$+ \frac{\gamma}{1-\gamma} \mathbb{E}_{(s,a,s') \sim d^{\pi^*}}[\lambda(s')(\tilde{V}^{\pi^*}(s') - h(s'))|s, s' \in \Omega]$$
$$+ \frac{\gamma}{1-\gamma} \mathbb{E}_{(s,a,s') \sim d^{\pi}}[\lambda(s')(h(s') - \tilde{V}^{\hat{\pi}}(s'))|s, s' \in \Omega].$$

Specifically, the performance of $\pi$ depends on both bias and regret. The bias term describes the discrepancy caused by solving the reshaped MDP with $\lambda(\cdot)$. The regret term describes the performance of the learned policy in the reshaped MDP.

To prove Theorem 3, we first conduct a regret decomposition as

$$V^*(s_0) - V^{\hat{\pi}}(s_0) = \left( V^*(s_0) - \tilde{V}^{\pi^*}(s_0) \right) + \left( \tilde{V}^{\pi^*}(s_0) - \tilde{V}^{\hat{\pi}}(s_0) \right) + \left( \tilde{V}^{\hat{\pi}}(s_0) - V^{\hat{\pi}}(s_0) \right).$$

Then, we rewrite both $V^*(s_0) - \tilde{V}^{\pi^*}(s_0)$ and $\tilde{V}^{\hat{\pi}}(s_0) - V^{\hat{\pi}}(s_0)$ using the heuristics $h(s)$.

### A.1    Technical Lemmas

Before proceeding to the proof details of Theorem 3, we first prove a lemma on $V^\pi(s_0) - \tilde{V}^\pi(s_0)$. For a policy $\pi$ under the original MDP $\mathcal{M}$ and the reshaped MDP $\tilde{\mathcal{M}}$, we aim to quantify the value difference using the heuristics $h(s)$.

**Lemma 4.** *For any policy* $\pi$*,*

$$V^\pi(s_0) - \tilde{V}^\pi(s_0) = \frac{\gamma}{1-\gamma} \mathbb{E}_{(s,a,s') \sim d^\pi}[\lambda(s')(\tilde{V}^\pi(s') - h(s'))|s, s' \in \Omega].$$

*Proof.* By the definition of $\lambda(s, s')$:

$$V^\pi(s_0) - \tilde{V}^\pi(s_0)$$
$$= \frac{1}{1-\gamma} \mathbb{E}_{(s,a,s') \sim d^\pi}[r(s,a) + \gamma \tilde{V}^\pi(s') - \tilde{V}^\pi(s)]$$
$$= \frac{1}{1-\gamma} \mathbb{E}_{(s,a,s') \sim d^\pi}[r(s,a) + \gamma \tilde{V}^\pi(s') - r(s,a) - \gamma\lambda(s,s')h(s') + \gamma(1 - \lambda(s,s'))\tilde{V}^\pi(s')]$$
$$= \frac{\gamma}{1-\gamma} \mathbb{E}_{(s,a,s') \sim d^\pi}[\lambda(s,s')(\tilde{V}^\pi(s') - h(s'))]$$
$$= \frac{\gamma}{1-\gamma} \mathbb{E}_{(s,a,s') \sim d^\pi}[\lambda(s,s')(\tilde{V}^\pi(s') - h(s'))|s, s' \in \Omega]$$
$$= \frac{\gamma}{1-\gamma} \mathbb{E}_{(s,a,s') \sim d^\pi}[\lambda(s')(\tilde{V}^\pi(s') - h(s'))|s, s' \in \Omega]$$

$\square$

### A.2    Proof of Theorem 3

To prove the theorem, we decompose the regret into the following terms:

$$V^*(s_0) - V^{\hat{\pi}}(s_0) = \left( V^*(s_0) - \tilde{V}^{\pi^*}(s_0) \right) + \left( \tilde{V}^{\pi^*}(s_0) - \tilde{V}^{\hat{\pi}}(s_0) \right) + \left( \tilde{V}^{\hat{\pi}}(s_0) - V^{\hat{\pi}}(s_0) \right)$$

We apply Lemma 4 to rewrite the first and the last terms as

$$V^*(s_0) - \tilde{V}^{\pi^*}(s_0) = \frac{\gamma}{1-\gamma} \mathbb{E}_{(s,a,s') \sim d^{\pi^*}}[\lambda(s')(\tilde{V}^{\pi^*}(s') - h(s'))|s, s' \in \Omega]$$
$$\tilde{V}^{\hat{\pi}}(s_0) - V^{\hat{\pi}}(s_0) = \frac{\gamma}{1-\gamma} \mathbb{E}_{(s,a,s') \sim d^{\pi}}[\lambda(s')(h(s') - \tilde{V}^{\hat{\pi}}(s'))|s, s' \in \Omega].$$

Combining the two completes the proof.

# B   EXTENDED RESULTS FOR THEOREM 2

To prove Theorem 2, we separately bound the bias term

$$\text{Bias}(\hat{\pi}, \lambda) := \frac{\gamma}{1-\gamma} \mathbb{E}_{(s,a,s') \sim d^{\pi^*}} [\lambda(s')(\tilde{V}^{\pi^*}(s') - h(s'))|s, s' \in \Omega],$$

and the regret term

$$\text{Regret}(\pi, h, \lambda) := \tilde{V}^{\pi^*}(d_0) - \tilde{V}^{\pi}(d_0) + \frac{\gamma}{1-\gamma} \mathbb{E}_{(s,a,s') \sim d^{\pi}} [\lambda(s')(h(s') - \tilde{V}^{\pi}(s'))|s, s' \in \Omega].$$

Specifically the bias is bounded by Lemma 6 and the regret is bounded by Lemma 9.

## B.1   TECHNICAL LEMMAS

Now, we provide some technical lemmas for the proof of Theorem 2.

### B.1.1   BIAS COMPONENT

We first prove the upper bound for the bias component.

**Lemma 5.** *Assume $h(s) \leq V^*(s)$, $\forall s \in \mathcal{S}$. It holds that $\tilde{V}^{\pi^*}(s) \leq V^*(s)$ for all $s \in \mathcal{S}$.*

*Proof.*

$$\begin{aligned}
\tilde{V}^{\pi^*}(s) &= r(s,a) + \gamma \mathbb{E}_{s'|s,a}[\lambda(s,s')h(s') + (1 - \lambda(s,s'))\tilde{V}^{\pi^*}(s')] \\
&= V^*(s) + \gamma \mathbb{E}_{s'|s,a}[\lambda(s,s')h(s') + (1 - \lambda(s,s'))\tilde{V}^{\pi^*}(s') - V^*(s')] \\
&= V^*(s) + \gamma \mathbb{E}_{s'|s,a}[\lambda(s,s')(h(s') - V^*(s')) + (1 - \lambda(s,s'))(\tilde{V}^{\pi^*}(s') - V^*(s'))] \\
&\leq V^*(s) + \gamma \mathbb{E}_{s'|s,a}[(1 - \lambda(s,s'))(\tilde{V}^{\pi^*}(s') - V^*(s'))]
\end{aligned}$$

where in the inequality we used $h(s) \leq V^{\pi^*}(s)$. Then by a contraction argument, we can show $\tilde{V}^{\pi^*}(s) - V^{\pi^*}(s) \leq 0$ $\qquad \square$

**Lemma 6** (Bias Upperbound). *Under the assumptions of Theorem 2, the bias component can be bounded by:*

$$\text{Bias}(\hat{\pi}, \lambda) \leq \frac{\lambda\gamma}{1-\gamma} \mathbb{E}_{(s,a,s') \sim d^{\pi^*}} [V^*(s') - V^{\mu}(s')|s, s' \in \Omega].$$

*Proof.* Review the bias component

$$\text{Bias}(\hat{\pi}, \lambda) := \frac{\gamma}{1-\gamma} \mathbb{E}_{(s,a,s') \sim d^{\pi^*}} [\lambda(s')(\tilde{V}^{\pi^*}(s') - h(s'))|s, s' \in \Omega].$$

Under the assumptions of Theorem 2, we derive

$$\text{Bias}(\hat{\pi}, \lambda) = \frac{\lambda\gamma}{1-\gamma} \mathbb{E}_{(s,a,s') \sim d^{\pi^*}} [\tilde{V}^{\pi^*}(s') - V^*(s') + V^*(s') - V^{\mu}(s')|s, s' \in \Omega].$$

Next by Lemma 5 we have $\frac{\lambda\gamma}{1-\gamma} \mathbb{E}_{(s,a,s') \sim d^{\pi^*}} [\tilde{V}^{\pi^*}(s') - V^*(s')|s, s' \in \Omega] \leq 0$, and thus

$$\text{Bias}(\hat{\pi}, \lambda) \leq \frac{\lambda\gamma}{1-\gamma} \mathbb{E}_{(s,a,s') \sim d^{\pi^*}} [V^*(s') - V^{\mu}(s')|s, s' \in \Omega].$$

$\qquad \square$

### B.1.2 REGRET COMPONENT

Next, we focus on the regret component.

**Lemma 7.** *Under the setup of Theorem 2, $\tilde{V}^\mu(s) = V^\mu(s)$.*

*Proof.* Since $\Omega$ is the support of $\mu$, this can be shown by the following: for $s \in \Omega$,

$$
\begin{aligned}
&\tilde{V}^\mu(s) - V^\mu(s) \\
&= \mathbb{E}_{a \sim \mu|s} \mathbb{E}_{s'|s,a}[r(s,a) + \gamma\lambda(s,s')h(s') + \gamma(1-\lambda(s,s'))\tilde{V}^\mu(s') - r(s,a) - \gamma V^\mu(s')] \\
&= \mathbb{E}_{a \sim \mu|s} \mathbb{E}_{s'|s,a}[r(s,a) + \gamma\lambda(s,s')h(s') + \gamma(1-\lambda(s,s'))\tilde{V}^\mu(s') - r(s,a) - \gamma V^\mu(s')|s' \in \Omega] \\
&= \mathbb{E}_{a \sim \mu|s} \mathbb{E}_{s'|s,a}[r(s,a) + \gamma\lambda(s,s')V^\mu(s') + \gamma(1-\lambda(s,s'))\tilde{V}^\mu(s') - r(s,a) - \gamma V^\mu(s')|s' \in \Omega] \\
&= \gamma\mathbb{E}_{a \sim \mu|s} \mathbb{E}_{s'|s,a}[(1-\lambda(s,s'))(\tilde{V}^\mu(s') - V^\mu(s'))|s' \in \Omega].
\end{aligned}
$$

Since $\gamma < 1$, then by an argument of contraction, we have $\tilde{V}^\mu(s) - V^\mu(s) = 0$ for $s \in \Omega$.

$\square$

**Lemma 8.** *The difference between $V^*(s)$ and $\tilde{V}^{\pi^*}(s)$ can be derived as*

$$
V^*(s) - \tilde{V}^{\pi^*}(s) = \mathbb{E}_{\rho^{\pi^*}(s)}[\sum_{t=1}^\infty [\lambda(1-\lambda)^{t-1}\gamma^t(V^*(s_t) - h(s_t))]].
$$

*Proof.* By the dynamic programming equation in both the original and shaped MDP.

$$
\begin{aligned}
V^*(s) - \tilde{V}^{\pi^*}(s) =& \gamma(1-\lambda)\mathbb{E}_{a \sim \pi^*(\cdot;s)}[\mathbb{E}_{s'|s,a}[V^*(s') - \tilde{V}^{\pi^*}(s')]] \\
&+ \gamma\lambda\mathbb{E}_{a \sim \pi^*(\cdot;s)}[\mathbb{E}_{s'|s,a}[V^*(s') - h(s')]].
\end{aligned}
\tag{6}
$$

Then, we use (6) recursively:

$$
V^*(s) - \tilde{V}^{\pi^*}(s) = \lambda\mathbb{E}_{\rho^{\pi^*}(s)}[\sum_{t=1}^\infty [(1-\lambda)^{t-1}\gamma^t(V^*(s_t) - h(s_t))]].
$$

$\square$

**Lemma 9** (Regret Upperbound). *The expected regret is bounded by*

$$
\mathbb{E}_\mathbb{D}[\text{Regret}(\hat{\pi}, \lambda(\cdot))] \lesssim \min\left(V_{max}, V_{max}\sqrt{\frac{(1-\gamma)|\mathcal{S}|\max_{s,a}\frac{d^{\pi^*}(s,a;d_0)}{\mu(s,a)}}{N(1-\gamma(1-\lambda))^4}}\right)
$$
$$
+ \frac{\gamma\lambda}{1-\gamma}\min\left(V_{max}, V_{max}\sqrt{\frac{(1-\gamma)|\mathcal{S}|\frac{1}{\min_{(s,a)\in\Omega}\mu(s,a)}}{N(1-\gamma(1-\lambda))^4}}\right).
$$

*Proof.* Under the setups in Theorem 2, we have

$$
\text{Regret}(\hat{\pi}, \lambda(\cdot)) = \tilde{V}^{\pi^*}(d_0) - \tilde{V}^{\hat{\pi}}(d_0) + \frac{\gamma\lambda}{1-\gamma}\mathbb{E}_{(s,a,s')\sim d^{\hat{\pi}}}[V^\mu(s') - \tilde{V}^{\hat{\pi}}(s')|s,s' \in \Omega].
$$

Further by Lemma 7, we can replace $V^\mu$ by $\tilde{V}^\mu$:

$$
\text{Regret}(\hat{\pi}, \lambda(\cdot)) = \tilde{V}^{\pi^*}(d_0) - \tilde{V}^{\hat{\pi}}(d_0) + \frac{\gamma\lambda}{1-\gamma}\mathbb{E}_{(s,a,s')\sim d^{\hat{\pi}}}[\tilde{V}^\mu(s') - \tilde{V}^{\hat{\pi}}(s')|s,s' \in \Omega].
\tag{7}
$$

Then, by Lemma 10,

$$
\mathbb{E}_\mathbb{D}[\tilde{V}^{\pi^*}(d_0) - \tilde{V}^{\hat{\pi}}(d_0)] \lesssim \min\left(V_{max}, V_{max}\sqrt{\frac{(1-\gamma)|\mathcal{S}|\max_{s,a}\frac{d^{\pi^*}(s,a;d_0)}{\mu(s,a)}}{N(1-\gamma(1-\lambda))^4}}\right),
\tag{8}
$$

and

$$\mathbb{E}_{\mathbb{D}}[\tilde{V}^\mu(d^{\hat\pi}) - \tilde{V}^{\hat\pi}(d^{\hat\pi})] \lesssim \min\left(V_{max}, V_{max}\sqrt{\frac{(1-\gamma)|\mathcal{S}|\max_{s,a}\frac{d^\mu(s,a;d^{\hat\pi})}{\mu(s,a)}}{N(1-\gamma(1-\lambda))^4}}\right).$$

Note that, by Lemma 11, $d^{\hat\pi}$ stays in $\Omega$. Therefore, we get $\max_{s,a}\frac{d^\mu(s,a;d^{\hat\pi})}{\mu(s,a)} = \max_{s,a\in\Omega}\frac{d^\mu(s,a;d^{\hat\pi})}{\mu(s,a)} \leq \frac{1}{\min_{(s,a)\in\Omega}\mu(s,a)}$, which leads to

$$\mathbb{E}_{\mathbb{D}}[\tilde{V}^\mu(d^{\hat\pi}) - \tilde{V}^{\hat\pi}(d^{\hat\pi})] \lesssim \min\left(V_{max}, V_{max}\sqrt{\frac{(1-\gamma)|\mathcal{S}|\frac{1}{\min_{(s,a)\in\Omega}\mu(s,a)}}{N(1-\gamma(1-\lambda))^4}}\right). \tag{9}$$

Take (8) and (9) into (7), we derive

$$\mathbb{E}_{\mathbb{D}}[\text{Regret}(\hat\pi, \lambda(\cdot))] \lesssim \min\left(V_{max}, V_{max}\sqrt{\frac{(1-\gamma)|\mathcal{S}|\max_{s,a}\frac{d^{\pi^*}(s,a;d_0)}{\mu(s,a)}}{N(1-\gamma(1-\lambda))^4}}\right)$$
$$+ \frac{\gamma\lambda}{1-\gamma}\min\left(V_{max}, V_{max}\sqrt{\frac{(1-\gamma)|\mathcal{S}|\frac{1}{\min_{(s,a)\in\Omega}\mu(s,a)}}{N(1-\gamma(1-\lambda))^4}}\right).$$

$\square$

## B.2 Proof of Theorem 2

The proof follows by combining Lemma 6 and 9.

## C VI-LCB with HUBL

We use the offline value iteration with lower confidence bound (VI-LCB) (Rashidinejad et al., 2021) as the base algorithm to analyze concretely the effects of HUBL with finite samples for the tabular case.

### C.1 Algorithm

We detail the procedure of HUBL when implemented with VI-LCB for the tabular setting. To start with, we introduce several definitions which will be used in the following algorithm and theoretical analysis. Without loss of generality, we assume that the sates take values in $\{1, 2, \cdots, |\mathcal{S}|\}$ and that the actions take values in $\{1, 2, \cdots, |\mathcal{A}|\}$. Then, let $h$ be a $|\mathcal{S}| \times 1$ vector which denotes the heuristic function. We assume each component satisfies $h_s = V^\mu(s)$ for $s \in \Omega$. Let $\Lambda$ be a $|\mathcal{S}| \times |\mathcal{S}|$ matrix with $\Lambda_{s,s'} = \lambda$ if $s, s' \in \Omega$ and $\Lambda_{s,s'} = 0$ if $s$ or $s' \notin \Omega$. We use $\odot$ to denote the component-wise multiplication, and $\Lambda_{s,:}$ to denote a row of $\Lambda$ as a $|\mathcal{S}| \times 1$ vector. With slight abuse of notation, we use $t$ to denote the index of iteration in this section, with $T$ as the total number of iterations.

To conduct VI-LCB with HUBL, we follow Algorithm 3 in Rashidinejad et al. (2021) but with a modified updating rule. The procedure is summarized Algorithm 2. Compared with the original VI-LCB, we highlight the key modification in Line 14 with blue color. Specifically, at the $t^{\text{th}}$ iteration, based on (2), we modify updating rule of Algorithm 3 in Rashidinejad et al. (2021) into

$$Q_t(s, a) \leftarrow r_t(s, a) - b_t(s, a) + \gamma P_{s,a}^t \odot (I - \Lambda_{s,:}) \cdot V_{t-1} + \gamma P_{s,a}^t \odot \Lambda_{s,:} \cdot h.$$

Note that we introduce heuristics by $\gamma P_{s,a}^t \odot \Lambda_{s,:} \cdot h$, while reducing the bootstrapping by $\gamma P_{s,a}^t \odot (I - \Lambda_{s,:}) \cdot V_{t-1}$.

### C.2 Regret Analysis

In this section, we study the regret under the reshaped MDP constructed by HUBL. Specifically, we bound the regret of Algorithm 2 in Lemma 10.

---

**Algorithm 2** HUBL with VI-LCB

---

1: **Input:** Batch dataset $\mathbb{D}$ and discount factor $\gamma$.
2: Set $T := \frac{\log N}{1-\gamma}$.
3: Randomly split $\mathbb{D}$ into $T + 1$ sets $\mathbb{D}_t = \{(s_i, a_i, r_i, s'_i)\}$ for $t \in \{0, 1, \cdots, T\}$, where $D_0$ consists of $\frac{N}{2}$ observations and other datasets have $\frac{N}{2T}$ observations.
4: Set $m_0(s, a) := \sum_{i=1}^{m} \mathbb{1}\{(s_i, a_i) = (s, a)\}$ based on dataset $\mathbb{D}_0$.
5: For all $a \in \mathcal{A}$ and $s \in \mathcal{S}$, initialize $Q_0(s, a) = 0$, $V_0(s) = 0$ and set $\pi_0(s) = \arg\max_a m_0(s, a)$.

6: **for** $t = 1, \cdots, T$ **do**
7:     Initialize $r_t(s, a) = 0$ and set $P^t_{s,a}$ to be a random probability vector.
8:     Set $m_t(s, a) := \sum_{i=1}^{m} \mathbb{1}\{(s_i, a_i) = (s, a)\}$ based on dataset $\mathbb{D}_t$.
9:     Compute penalty $b_t(s, a)$ for $L = 2000 \log(2(T + 1)|\mathcal{S}||\mathcal{A}|N)$

$$b_t(s, a) := V_{\max} \sqrt{\frac{L}{m_t(s, a) \vee 1}}.$$

10:     **for** $(s, a) \in (\mathcal{S}, \mathcal{A})$ **do**
11:       **if** $m_t(s, a) \geq 1$ **then**
12:         Set $P^t_{s,a}$ to be empirical transitions and $r_t(s, a)$ be empirical average of rewards.
13:         Set $Q_t(s, a) \leftarrow r_t(s, a) - b_t(s, a) + \gamma P^t_{s,a} \odot (I - \Lambda_{s,:}) \cdot V_{t-1} + \gamma P^t_{s,a} \odot \Lambda_{s,:} \cdot h$.
14:     Compute $V^{mid}_t(s) \leftarrow \max_a Q_t(s, a)$ and $\pi^{mid}_t(s) \in \arg\max_a Q_t(s, a)$.
15:     **for** $s \in \mathcal{S}$ **do**
16:       **if** $V^{mid}_t(s) \leq V_{t-1}(s)$ **then**
17:         $V_t(s) \leftarrow V_{t-1}(s)$ and $\pi_t(s) \leftarrow \pi_{t-1}(s)$.
18:       **else**
19:         $V_t(s) \leftarrow V^{mid}_t(s)$ and $\pi_t(s) \leftarrow \pi^{mid}_t(s)$.
20: **Return** $\hat{\pi} := \pi_T$.

---

**Lemma 10** (Regret of VI-LCB with HUBL). *Let the assumptions in Section C.3 be satisfied. Then, for any initial distribution $d_{init}$ in $\Omega$, the regret of Algorithm 2 is bounded by*

$$\mathbb{E}_{\mathbb{D}}[\tilde{V}^{\pi^*}(d_{init}) - \tilde{V}^{\hat{\pi}}(d_{init})] \lesssim \min\left(V_{max}, V_{max}\sqrt{\frac{(1-\gamma)|\mathcal{S}|\max_{s,a}\frac{d^{\pi^*}(s,a;d_{init})}{\mu(s,a)}}{N(1-\gamma(1-\lambda))^4}}\right).$$

### C.3 NOTATIONS

We first provide some matrix notations for MDPs. We use $P^\pi \in \mathbb{R}^{|\mathcal{S}||\mathcal{A}|\times|\mathcal{S}||\mathcal{A}|}$ to denote the transition matrix induced by policy $\pi$ whose $(s, a) \times (s, a')$ element is equal to $P(s'|s, a)\pi(a'|s')$, and $d^\pi \in \mathbb{R}^{|\mathcal{S}||\mathcal{A}|}$ to denote a state-action distribution induced by policy $\pi$ whose $(s, a)$ element is equal to $d(s)\pi(a|s)$. Similarly, we use $\Lambda_Q \in \mathbb{R}^{|\mathcal{S}||\mathcal{A}|\times|\mathcal{S}||\mathcal{A}|}$ to denote a extended matrix version of $\Lambda$.

Further, we focus on the policies which stay in the data distribution support $\Omega$. Such polices are formally defined as $\{\pi|d^\pi(s, a; s_0) = 0$ for any $(s, a) \notin \Omega$ and $s_0 \in \Omega\}$. We also define a clean event:

$$\mathcal{E}_{MDP} := \left\{\forall s, a, t, \left|r(s, a) - r_t(s, a) + \gamma(P_{s,a} - P^t_{s,a}) \odot (I - \Lambda) \cdot V_{t-1}\right| \leq b_t(s, a)\right\}.$$

### C.4 TECHNICAL LEMMAS

With the aforementioned definitions and assumptions, we provide the following lemmas.

**Lemma 11.** *Let $\pi$ be a policy learned by Algorithm 2. Under the event $\mathcal{E}_{cover} := \{\Omega \subseteq \mathbb{D}_0\}$ $\pi$ stays in $\Omega$ for any initial state in $\Omega$. The probability of $\mathcal{E}_{cover}$ is greater than*

$$1 - |\mathcal{S}|(1 - \min_{s\in\Omega}\mu(s))^{\frac{N}{2}}.$$

*Proof.* Under the event $\mathcal{E}_{cover}$, we first fix $s \in \Omega$ and show that given $s$, the learned policy $\pi$ does not take any action $a' \notin \Omega$. By Algorithm 2, for any $(s, a') \notin \Omega$ and $t = 1, 2, \cdots, T$, we can derive

$$Q_t(s, a') \leq -V_{\max}\sqrt{2000 \log(2T+1)|\mathcal{S}||\mathcal{A}|N} + \gamma V_{\max}.$$

Since $N$ and $T$ are larger than 1, we can conclude that

$$Q_t(s, a') \leq 0 \text{ for any } a' \text{ such that } (s, a') \notin \Omega.$$

Next we consider two cases:

*Case I:* If there exists $a$ such that $(s, a) \in \Omega$ and that there exists $t = 1, 2, \cdots, T$ such that

$$Q_t(s, a) > 0,$$

we can conclude that $Q_T(s, a) \geq Q_t(s, a) > Q(s, a')$, which suggests that the learned policy $\pi$ never conducts action $a' \notin \Omega$ given state $s$.

*Case II:* If for any $a$ such that $(s, a) \in \Omega$ and any $t = 1, 2, \cdots, T$, we have

$$Q_t(s, a) = 0,$$

according to Algorithm 2, $\pi(s) = \arg\max_a m_0(s, a)$. By the definition of $\mathcal{E}_{cover}$, the policy $\pi$ stays in $\Omega$.

Next, we consider the probability of $\mathcal{E}_{cover}$. Given a state $s \in \Omega$, we define the event $\mathcal{E}_s := \{m_0(s, a) = 0 \text{ for all } a \text{ such that } (s, a) \in \Omega\}$. The probability of $\mathcal{E}_s$ can be derive as

$$P(\mathcal{E}_s) \leq (1 - \mu(s))^{\frac{N}{2}}.$$

By union bound, we can derive that

$$P(\neg\mathcal{E}_{cover}) \leq |\mathcal{S}|(1 - \min_{s\in\Omega} \mu(s))^{\frac{N}{2}}.$$

As a result, $\pi$ stays in $\Omega$ with the probability greater than

$$1 - |\mathcal{S}|(1 - \min_{s\in\Omega} \mu(s))^{\frac{N}{2}}.$$

$\square$

**Lemma 12.** *Let $\pi$ be a policy that stays in $\Omega$. Then, with $d_{init}$ as any initial distribution in $\Omega$, we can derive*

$$\frac{d_{init}(s)}{\mu(s, \pi(s))} \leq \frac{\frac{\tilde{d}^\pi(s, \pi(s); d_{init})}{\mu(s, \pi(s))}}{1 - \gamma(1 - \lambda)}.$$

*Proof.* By definition, we have

$$\tilde{d}^\pi(s, \pi(s); d_{init}(s)) := \frac{1}{1 - \gamma(1 - \lambda)} \sum_{t=0}^{\infty} \gamma^t (1 - \lambda)^t P_t(S_t = s, \pi; d_{init})$$

Therefore, $d_{init}(s) \leq \frac{1}{1-\gamma(1-\lambda)}\tilde{d}^\pi(s, \pi(s); d_{init})$, which finishes the proof.

$\square$

**Lemma 13.** *Let $v_k^\pi = d_{init}^\pi(\gamma P^\pi \odot (I - \Lambda_Q))^k$. For a policy $\pi$ that stays in $\Omega$, the following equality holds: $v_k^\pi = d_{init}^\pi(\gamma(1 - \lambda)P^\pi)^k$.*

*Proof.* Since $\pi$ stays in $\Omega$, $P^\pi$ only accesses the entries in $I - \Lambda_Q$ whose values equal to $(1 - \lambda)$. This observation finishes the proof. $\square$

**Lemma 14.** *Let $\pi$ be a policy that stays in $\Omega$. Under the event $\mathcal{E}_{MDP}$, for all $t = 1, 2, \cdots, T$,*

$$\tilde{V}(\pi) - \tilde{V}(\pi_t) \leq V_{max}\gamma^t(1 - \lambda)^t + 2\sum_{i=1}^{t} \mathbb{E}_{v_{t-i}^\pi}[b_i(s, a)].$$

*Proof.* The proof follows the Lemma 2 of Rashidinejad et al. (2021) combined with Lemma 13. □

**Lemma 15.** *For any policy $\pi$ that stays in $\Omega$, the following inequality is true:*

$$\tilde{d}^\pi(s,a;d_{init}) \le \frac{1-\gamma}{1-\gamma(1-\lambda)} d^\pi(s,a;d_{init}),$$

*for any $(s,a) \in \Omega$.*

*Proof.* By definition, we have

$$\tilde{d}^\pi(s,a;d_{init}) := \frac{1}{1-\gamma(1-\lambda)} \sum_{t=0}^{\infty} \gamma^t (1-\lambda)^t \mathrm{P}_t(S_t = s, A_t = a, ; \pi, d_{init})$$

$$d^\pi(s,a;d_{init}) := \frac{1}{1-\gamma} \sum_{t=0}^{\infty} \gamma^t \mathrm{P}_t(S_t = s, A_t = a; \pi, d_{init}).$$

Therefore,

$$\tilde{d}^\pi(s,a;d_{init}) \le \frac{1}{1-\gamma(1-\lambda)} \sum_{t=0}^{\infty} \gamma^t \mathrm{P}_t(S_t = s, A_t = a; \pi, d_{init}) = \frac{1-\gamma}{1-\gamma(1-\lambda)} d^\pi(s,a;d_{init})$$

□

## C.5 PROOF OF LEMMA 10

By the event $\mathcal{E}_{cover}$, we can decompose the regret into

$$\mathbb{E}_{\mathbb{D}}[\tilde{V}^{\pi^*}(d_{init}) - \tilde{V}^{\hat{\pi}}(d_{init})]$$
$$\le \mathbb{E}_{\mathbb{D}}[\tilde{V}^{\pi^*}(d_{init}) - \tilde{V}^{\hat{\pi}}(d_{init}) \mathbb{1}\{\mathcal{E}_{cover}\}] + \mathbb{E}_{\mathbb{D}}[\tilde{V}^{\pi^*}(d_{init}) - \tilde{V}^{\hat{\pi}}(d_{init}) \mathbb{1}\{\mathcal{E}^c_{cover}\}] \quad (10)$$
$$\le \mathbb{E}_{\mathbb{D}}[\tilde{V}^{\pi^*}(d_{init}) - \tilde{V}^{\hat{\pi}}(d_{init}) \mathbb{1}\{\mathcal{E}_{cover}\}] + V_{max}|\mathcal{S}| \left(\frac{|\mathcal{S}|-1}{|\mathcal{S}|}\right)^{\frac{N}{2}},$$

where the second inequality is by Lemma 11. Next, we focus on $\mathbb{E}_{\mathbb{D}}[\tilde{V}^{\pi^*}(d_{init}) - \tilde{V}^{\hat{\pi}}(d_{init}) \mathbb{1}\{\mathcal{E}_{cover}\}]$. Following the analysis in C.5 of Rashidinejad et al. (2021), we can derive

$$\mathbb{E}_{\mathbb{D}}[(\tilde{V}^{\pi^*}(d_{init}) - \tilde{V}^{\hat{\pi}}(d_{init})) \mathbb{1}\{\mathcal{E}_{cover}\}]$$
$$\le \mathbb{E}_{\mathbb{D}}[(\tilde{V}^{\pi^*}(d_{init}) - \tilde{V}^{\hat{\pi}}(d_0)) \mathbb{1}\{\mathcal{E}_{cover}\}]$$
$$\le \mathbb{E}_{\mathbb{D}}[\mathbb{E}_{s \sim d_{init}}[(\tilde{V}^{\pi^*}(d_{init}) - \tilde{V}^{\hat{\pi}}(d_{init})) \mathbb{1}\{\exists t \le T, m_t(s, \pi^*(s)) = 0\} \mathbb{1}\{\mathcal{E}_{cover}\}]] := T_1$$
$$+ \mathbb{E}_{\mathbb{D}}[\mathbb{E}_{s \sim d_{init}}[(\tilde{V}^{\pi^*}(d_{init}) - \tilde{V}^{\hat{\pi}}(d_{init}))$$
$$\mathbb{1}\{\forall t \le T, m_t(s, \pi^*(s)) \ge 1\} \mathbb{1}\{\mathcal{E}_{MDP}\} \mathbb{1}\{\mathcal{E}_{cover}\}]] := T_2$$
$$+ \mathbb{E}_{\mathbb{D}}[\mathbb{E}_{s \sim d_{init}}[(\tilde{V}^{\pi^*}(d_{init}) - \tilde{V}^{\hat{\pi}}(d_{init}))$$
$$\mathbb{1}\{\forall t \le T, m_t(s, \pi^*(s)) \ge 1\} \mathbb{1}\{\mathcal{E}^c_{MDP}\} \mathbb{1}\{\mathcal{E}_{cover}\}]] := T_3.$$

By Lemma 11, under the event $\mathcal{E}_{cover}$, $\hat{\pi}$ stays in $\Omega$. In other words, Lemma 12, 13 and 14 are all applicable to $\hat{\pi}$.

Next, for the following analysis, we consider the case that $\pi^*$ stays in $\Omega$. We will discuss the case that $\Omega$ does not cover $\pi^*$ at the end of the proof.

By Lemma 12 and C.5.1 of Rashidinejad et al. (2021),

$$T_1 \le \frac{8 V_{max} |\mathcal{S}| T^2 \max_{s,a} \frac{\tilde{d}^{\pi^*}(s,a;d_{init})}{\mu(s,a)}}{9(1-(1-\lambda)\gamma)N}.$$

By Lemma 14 and C.5.2 of Rashidinejad et al. (2021),

$$T_2 \le V_{max} \gamma^T (1-\lambda)^T + 32 \frac{V_{max}}{1-\gamma(1-\lambda)} \sqrt{\frac{2L|\mathcal{S}|T \max_{s,a} \frac{\tilde{d}^{\pi^*}(s,a;d_{init})}{\mu(s,a)}}{N}}.$$

By C.5.2 of Rashidinejad et al. (2021),

$$T_3 \leq \frac{V_{max}}{N}.$$

Combining $T_1$, $T_2$ $T_3$, and (10), with $T = \log N/(1 - \gamma(1 - \lambda))$,

$$\mathbb{E}_\mathbb{D}[\tilde{V}^{\pi^*}(d_{init}) - \tilde{V}^{\hat{\pi}}(d_{init})] \lesssim \min\left(V_{max}, V_{max}\sqrt{\frac{|\mathcal{S}|\max_{s,a}\frac{\tilde{d}^{\pi^*}(s,a;d_{init})}{\mu(s,a)}}{N(1 - \gamma(1 - \lambda))^3}}\right).$$

Finally, by Lemma 15, we finish the proof:

$$\mathbb{E}_\mathbb{D}[\tilde{V}^{\pi^*}(d_{init}) - \tilde{V}^{\hat{\pi}}(d_{init})] \lesssim \min\left(V_{max}, V_{max}\sqrt{\frac{(1-\gamma)|\mathcal{S}|\max_{s,a}\frac{d^{\pi^*}(s,a;d_{init})}{\mu(s,a)}}{N(1 - \gamma(1 - \lambda))^4}}\right). \quad (11)$$

Note that (11) is derived under the assumption that $\pi^*$ stays in $\Omega$. However, it also holds when $\pi^*$ is not covered by $\Omega$. In that case, $\max_{s,a}\frac{d^{\pi^*}(s,a;d_{init})}{\mu(s,a)} = \infty$ and $\mathbb{E}_\mathbb{D}[\tilde{V}^{\pi^*}(d_{init}) - \tilde{V}^{\hat{\pi}}(d_{init})] \leq V_{max}$.

## D EXTENDED RESULTS FOR EXPERIMENTS

We conduct HUBL with four offline RL base methods on 27 datasets in D4RL and Meta-World. By blending heuristics with bootstrapping, HUBL reduces the complexity of decision-making and provides smaller regret while generating limited bias. We demonstrate that HUBL is able to improve the performance of offline RL methods.

### D.1 IMPLEMENTATION DETAILS

We implement base offline RL methods with code sources provided in Table 3.

Table 3: Code source for base methods.

| Base Methods | Code Source |
|---|---|
| ATAC | `https://github.com/chinganc/lightATAC` |
| CQL | `https://github.com/young-geng/CQL/tree/master/SimpleSAC` |
| IQL | `https://github.com/gwthomas/IQL-PyTorch/blob/main/README.md` |
| TD3+BC | `https://github.com/sfujim/TD3_BC` |

Experiments with ATAC, IQL, and TD3+BC are ran on Standard_F4S_V2 nodes of Azure, and experiments with CQL are ran on NC6S_V2 nodes of Azure. As suggested by the original implementation of the considered base offline RL methods, we use 3-layer fully connected neural networks for policy critics and value networks, where each hidden layer has 256 neurons and ReLU activation and the output layer is linear. The first-order optimization is implemented by ADAM (Kingma and Ba, 2014) with a minibatch size as 256. The learning rates are selected following the original implementation and are reported in Table 4.

Table 4: Learning rates

| Base Methods | Policy Network | Q Network |
|---|---|---|
| ATAC | $5 \times 10^{-7}$ | $5 \times 10^{-4}$ |
| CQL | $3 \times 10^{-4}$ | $3 \times 10^{-4}$ |
| IQL | $3 \times 10^{-4}$ | $3 \times 10^{-4}$ |
| TD3_BC | $3 \times 10^{-4}$ | $3 \times 10^{-4}$ |

For each dataset, the hyperparameters of base methods are tuned from six different configurations suggested by the original papers. Such configurations are summarized in Table 5.

Table 5: Learning rates

| Base Methods | Hyperparameter | Values |
|---|---|---|
| ATAC | degree of pessimism $\beta$ | $\{1.0, 4.0, 16.0, 0.25, 0.625, 10\}$ |
| CQL | min Q weight | $\{5, 20, 80, 1.25, 0.3125, 10.0\}$ |
| IQL | inverse temperature $\beta$ | $\{6.5, 3.0, 10.0\}$ |
| | expectile parameter $\tau$ | $\{0.7, 0.9\}$ |
| TD3+BC | strength of the regularizer $\lambda$ | $\{2.5, 0.625, 10.0, 40.0, 0.15625, 1\}$ |

Meanwhile, HUBL has one extra hyperparameter, $\alpha$, which is *fixed* for all the datasets but different for each base method. Specifically, $\alpha$ is selected from $\{0.001, 0.01, 0.1\}$ according to the relative improvement averaged over all the datasets in one task. Later in the robustness analysis, we show that the performance of HUBL is insensitive to the selection of $\alpha$. For each configuration and each base method, we repeat experiments three times with seeds in $\{0, 1, 10\}$.

## D.2 BASE PERFORMANCE AND STANDARD DEVIATIONS FOR D4RL DATASETS

We provide the performance of base offline RL methods and standarde deviations in Table 6.

Table 6: Relative normalized score improvement for HUBL and normalized scores of base methods on D4RL datasets.

| | Sigmoid | Rank | Constant | ATAC |
|---|---|---|---|---|
| halfcheetah-medium-expert-v2 | **0.02 ± 0.0** | **0.02 ± 0.01** | **0.02 ± 0.0** | 92.5 ± 0.73 |
| halfcheetah-medium-replay-v2 | 0.01 ± 0.01 | **0.02 ± 0.01** | **0.02 ± 0.02** | 46.89 ± 0.25 |
| halfcheetah-medium-v2 | −0.0 ± 0.01 | 0.0 ± 0.0 | **0.02 ± 0.01** | 52.54 ± 0.63 |
| hopper-medium-expert-v2 | 0.0 ± 0.0 | 0.0 ± 0.0 | 0.0 ± 0.0 | 111.04 ± 0.13 |
| hopper-medium-replay-v2 | 0.0 ± 0.0 | 0.0 ± 0.01 | 0.0 ± 0.0 | 101.23 ± 0.59 |
| hopper-medium-v2 | 0.01 ± 0.04 | **0.02 ± 0.07** | −0.03 ± 0.15 | 85.59 ± 4.76 |
| walker2d-medium-expert-v2 | 0.0 ± 0.0 | **0.02 ± 0.0** | 0.0 ± 0.01 | 111.99 ± 0.8 |
| walker2d-medium-replay-v2 | 0.03 ± 0.03 | **0.08 ± 0.0** | 0.05 ± 0.08 | 84.22 ± 2.7 |
| walker2d-medium-v2 | 0.01 ± 0.01 | −0.02 ± 0.05 | **0.02 ± 0.0** | 87.2 ± 0.67 |
| Average Relative Improvement | 0.01 | 0.02 | 0.01 | - |
| | Sigmoid | Rank | Constant | CQL |
| halfcheetah-medium-expert-v2 | 0.02 ± 0.02 | **0.03 ± 0.06** | −0.02 ± 0.17 | 81.96 ± 9.93 |
| halfcheetah-medium-replay-v2 | 0.1 ± 0.01 | **0.18 ± 0.02** | 0.13 ± 0.01 | 40.98 ± 0.81 |
| halfcheetah-medium-v2 | −0.1 ± 0.01 | **−0.02 ± 0.01** | −0.03 ± 0.01 | 51.05 ± 0.83 |
| hopper-medium-expert-v2 | **0.42 ± 0.01** | 0.39 ± 0.07 | 0.38 ± 0.05 | 78.39 ± 14.51 |
| hopper-medium-replay-v2 | **0.03 ± 0.01** | 0.01 ± 0.01 | 0.02 ± 0.01 | 97.21 ± 3.15 |
| hopper-medium-v2 | **0.18 ± 0.14** | 0.15 ± 0.17 | 0.17 ± 0.06 | 70.27 ± 8.77 |
| walker2d-medium-expert-v2 | 0.21 ± 0.0 | 0.22 ± 0.01 | **0.22 ± 0.0** | 89.38 ± 17.54 |
| walker2d-medium-replay-v2 | 0.08 ± 0.27 | **0.38 ± 0.06** | 0.37 ± 0.04 | 63.35 ± 6.22 |
| walker2d-medium-v2 | 0.05 ± 0.05 | **0.05 ± 0.03** | −0.03 ± 0.15 | 79.41 ± 1.82 |
| Average Relative Improvement | 0.11 | 0.16 | 0.13 | - |
| | Sigmoid | Rank | Constant | IQL |
| halfcheetah-medium-expert-v2 | 0.0 ± 0.01 | **0.0 ± 0.0** | −0.05 ± 0.03 | 92.92 ± 0.5 |
| halfcheetah-medium-replay-v2 | −0.03 ± 0.02 | **0.0 ± 0.02** | −0.04 ± 0.03 | 44.35 ± 0.21 |
| halfcheetah-medium-v2 | −0.01 ± 0.0 | **0.0 ± 0.0** | −0.01 ± 0.01 | 49.61 ± 0.16 |
| hopper-medium-expert-v2 | −0.04 ± 0.03 | **−0.03 ± 0.02** | −0.09 ± 0.12 | 108.13 ± 2.69 |
| hopper-medium-replay-v2 | **0.72 ± 0.09** | 0.58 ± 0.25 | 0.62 ± 0.27 | 53.98 ± 4.18 |
| hopper-medium-v2 | **0.03 ± 0.03** | 0.02 ± 0.03 | −0.01 ± 0.15 | 61.03 ± 3.27 |
| walker2d-medium-expert-v2 | **0.01 ± 0.0** | 0.01 ± 0.01 | **0.01 ± 0.0** | 109.59 ± 0.2 |
| walker2d-medium-replay-v2 | 0.15 ± 0.12 | 0.21 ± 0.02 | **0.13 ± 0.04** | 73.57 ± 2.64 |
| walker2d-medium-v2 | 0.03 ± 0.02 | **0.03 ± 0.01** | 0.04 ± 0.01 | 81.68 ± 2.48 |
| Average Relative Improvement | 0.09 | 0.09 | 0.06 | - |
| | Sigmoid | Rank | Constant | TD3+BC |
| halfcheetah-medium-expert-v2 | −0.02 ± 0.03 | −0.02 ± 0.03 | **−0.02 ± 0.02** | 94.82 ± 0.33 |
| halfcheetah-medium-replay-v2 | −0.1 ± 0.03 | −0.08 ± 0.16 | **−0.0 ± 0.02** | 54.38 ± 1.24 |
| halfcheetah-medium-v2 | −0.07 ± 0.02 | −0.04 ± 0.01 | **−0.01 ± 0.01** | 64.63 ± 2.15 |
| hopper-medium-expert-v2 | −0.06 ± 0.13 | −0.16 ± 0.2 | **0.03 ± 0.05** | 98.79 ± 12.57 |
| hopper-medium-replay-v2 | 0.16 ± 0.08 | 0.21 ± 0.03 | **0.24 ± 0.03** | 81.43 ± 27.77 |
| hopper-medium-v2 | 0.51 ± 0.17 | **0.7 ± 0.02** | 0.6 ± 0.19 | 59.12 ± 3.39 |
| walker2d-medium-expert-v2 | 0.0 ± 0.01 | **0.01 ± 0.01** | 0.0 ± 0.0 | 111.87 ± 0.17 |
| walker2d-medium-replay-v2 | **0.12 ± 0.04** | 0.09 ± 0.07 | 0.05 ± 0.02 | 84.95 ± 1.85 |
| walker2d-medium-v2 | −0.01 ± 0.03 | **0.08 ± 0.01** | 0.0 ± 0.07 | 84.09 ± 1.94 |
| Average Relative Improvement | 0.06 | 0.09 | 0.1 | - |

Next, we study how the performance of HUBL vary over different epochs. Specifically, we provide a plot of the achieved normalized average returns over different epochs for HUBL with TD3+BC on hopper-medium-v2 in Figure 4.

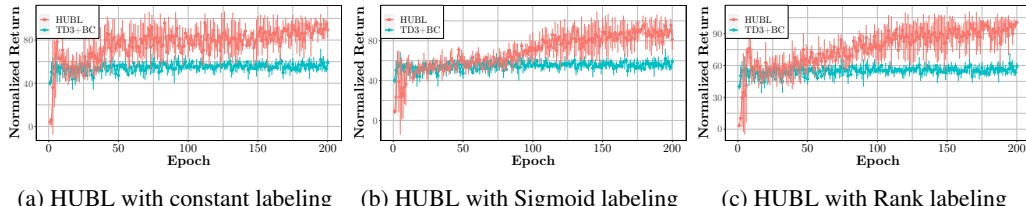

(a) HUBL with constant labeling  (b) HUBL with Sigmoid labeling  (c) HUBL with Rank labeling

Figure 4: Average normalized return of HUBL with TD3+BC on hopper-medium-v2

### D.3 ROBUSTNESS OF HUBL TO $\alpha$

In this section, we demonstrate the robustness of HUBL to $\alpha$. Specifically, we provide the average relative improvements of HUBL on D4RL datasets in Table 7. Notice that most average relative improvements are positive, which shows that HUBL can improve the performance with different values.

Table 7: Average relative improvements of HUBL under different $\alpha$'s

|  | Sigmoid | Rank | Constant |
|---|---|---|---|
| $\alpha = 0.001$ | 0.01 | 0.01 | 0 |
| $\alpha = 0.01$ | 0.01 | 0.01 | 0.01 |
| $\alpha = 0.1$ | 0 | 0.02 | 0.01 |

(a) ATAC

|  | Sigmoid | Rank | Constant |
|---|---|---|---|
| $\alpha = 0.001$ | 0.05 | 0.05 | 0.03 |
| $\alpha = 0.01$ | 0.08 | 0.05 | 0.10 |
| $\alpha = 0.1$ | 0.11 | 0.16 | 0.13 |

(b) CQL

|  | Sigmoid | Rank | Constant |
|---|---|---|---|
| $\alpha = 0.001$ | 0.02 | 0 | -0.04 |
| $\alpha = 0.01$ | 0 | -0.01 | -0.02 |
| $\alpha = 0.1$ | 0.09 | 0.09 | 0.06 |

(c) IQL

|  | Sigmoid | Rank | Constant |
|---|---|---|---|
| $\alpha = 0.001$ | 0.01 | 0.01 | 0 |
| $\alpha = 0.01$ | 0.01 | 0.01 | 0.01 |
| $\alpha = 0.1$ | 0.06 | 0.09 | 0.06 |

(d) TD3_BC

The selected $\alpha$'s are reported in Table 8:

Table 8: Selected $\alpha$'s

|  | Sigmoid | Rank | Constant |
|---|---|---|---|
| ATAC | 0.01 | 0.01 | 0.1 |
| CQL | 0.1 | 0.1 | 0.1 |
| IQL | 0.1 | 0.1 | 0.1 |
| TD3_BC | 0.01 | 0.1 | 0.1 |

## D.4 Data Collection for Meta-World

We collect data for Meta-World tasks using normalized rewards with goal-oriented stopping. Specifically, given that the original rewards of Meta-World are in $[0, 10]$, we shift them by $-10$ and divide by 10, so that the normalized rewards take values in $[-1, 0]$. Then, we use the hand scripted policy given in Meta-World with different Gaussian noise levels in $\{0.1, 0.5, 1\}$ to collect 100 trajectory for each task. In this process, a trajectory ends if (i) it reaches the max length of a trajectory (150); (ii) it finishes the goal. Note that we follow the same rule when testing the performance of a learned policy.

## D.5 Base Performance and Standard Deviations for Meta-World Datasets

We provide the performance of base offline RL methods and standard deviations in Table 9 and 10. We also consider behavior cloning (BC) as as a baseline, and also a representative for imitation learning and inverse reinforcement learning methods (Geng et al., 2020; 2023; Fu et al., 2017). Since the behavior policy is the scripted policy with Gaussian noise, BC can effective recover the scripted policy and thus is especially competitive.

Table 9: Relative normalized score improvement for HUBL and normalized scores of base methods on MW datasets (part I)

| | Sigmoid | Rank | Constant | ATAC | BC |
|---|---|---|---|---|---|
| button-press-v2–noise0.1 | $-0.0 \pm 0.03$ | $0.01 \pm 0.01$ | $\mathbf{0.01 \pm 0.01}$ | $-58.12 \pm 0.45$ | -58.79 |
| button-press-v2–noise0.5 | $-0.0 \pm 0.03$ | $-0.03 \pm 0.05$ | $\mathbf{0.01 \pm 0.04}$ | $-64.16 \pm 0.93$ | -59.58 |
| button-press-v2–noise1 | $-0.05 \pm 0.07$ | $\mathbf{-0.0 \pm 0.04}$ | $-0.01 \pm 0.04$ | $-62.72 \pm 3.84$ | -61.24 |
| push-back-v2–noise0.1 | $0.1 \pm 0.03$ | $0.04 \pm 0.07$ | $\mathbf{0.15 \pm 0.06}$ | $-87.12 \pm 13.89$ | -104.85 |
| push-back-v2–noise0.5 | $0.11 \pm 0.07$ | $\mathbf{0.19 \pm 0.05}$ | $0.17 \pm 0.02$ | $-133.28 \pm 14.24$ | -134.63 |
| push-back-v2–noise1 | $0.04 \pm 0.05$ | $0.02 \pm 0.09$ | $\mathbf{0.06 \pm 0.07}$ | $-141.29 \pm 13.87$ | -143.63 |
| reach-v2–noise0.1 | $-0.0 \pm 0.06$ | $-0.0 \pm 0.05$ | $\mathbf{0.0 \pm 0.07}$ | $-17.19 \pm 0.76$ | -18.31 |
| reach-v2–noise0.5 | $0.37 \pm 0.05$ | $\mathbf{0.46 \pm 0.06}$ | $0.43 \pm 0.08$ | $-33.91 \pm 5.62$ | -24.57 |
| reach-v2–noise1 | $0.39 \pm 0.05$ | $0.35 \pm 0.24$ | $\mathbf{0.5 \pm 0.05}$ | $-46.58 \pm 3.98$ | -43.79 |
| Average Relative Improvement | 0.11 | 0.11 | 0.14 | - | - |
| | Sigmoid | Rank | Constant | CQL | BC |
| button-press-v2–noise0.1 | $0.0 \pm 0.03$ | $0.02 \pm 0.05$ | $\mathbf{0.03 \pm 0.03}$ | $-59.67 \pm 1.69$ | -58.79 |
| button-press-v2–noise0.5 | $-0.04 \pm 0.05$ | $-0.02 \pm 0.01$ | $\mathbf{-0.02 \pm 0.0}$ | $-58.48 \pm 0.85$ | -59.58 |
| button-press-v2–noise1 | $-0.08 \pm 0.12$ | $\mathbf{0.08 \pm 0.02}$ | $0.04 \pm 0.11$ | $-67.18 \pm 14.25$ | -61.24 |
| push-back-v2–noise0.1 | $-0.07 \pm 0.07$ | $0.02 \pm 0.05$ | $\mathbf{0.02 \pm 0.02}$ | $-81.78 \pm 2.88$ | -104.85 |
| push-back-v2–noise0.5 | $\mathbf{0.25 \pm 0.12}$ | $0.22 \pm 0.1$ | $0.19 \pm 0.12$ | $-124.64 \pm 2.45$ | -134.63 |
| push-back-v2–noise1 | $0.06 \pm 0.13$ | $\mathbf{0.11 \pm 0.14}$ | $0.1 \pm 0.05$ | $-133.66 \pm 8.17$ | -143.63 |
| reach-v2–noise0.1 | $-0.04 \pm 0.17$ | $\mathbf{-0.01 \pm 0.05}$ | $-0.04 \pm 0.02$ | $-18.5 \pm 2.52$ | -18.31 |
| reach-v2–noise0.5 | $\mathbf{0.51 \pm 0.05}$ | $0.18 \pm 0.53$ | $0.4 \pm 0.27$ | $-41.02 \pm 18.79$ | -24.57 |
| reach-v2–noise1 | $0.8 \pm 0.02$ | $0.35 \pm 0.15$ | $\mathbf{0.81 \pm 0.03}$ | $-107.14 \pm 32.75$ | -43.79 |
| Average Relative Improvement | 0.16 | 0.1 | 0.17 | - | - |
| | Sigmoid | Rank | Constant | IQL | BC |
| button-press-v2–noise0.1 | $0.02 \pm 0.02$ | $0.03 \pm 0.01$ | $\mathbf{0.03 \pm 0.01}$ | $-59.87 \pm 1.32$ | -58.79 |
| button-press-v2–noise0.5 | $-0.01 \pm 0.05$ | $\mathbf{-0.01 \pm 0.02}$ | $-0.14 \pm 0.13$ | $-60.68 \pm 0.29$ | -59.58 |
| button-press-v2–noise1 | $-0.09 \pm 0.15$ | $\mathbf{0.07 \pm 0.01}$ | $0.04 \pm 0.03$ | $-73.96 \pm 5.09$ | -61.24 |
| push-back-v2–noise0.1 | $0.01 \pm 0.04$ | $\mathbf{0.04 \pm 0.02}$ | $0.03 \pm 0.01$ | $-83.79 \pm 5.45$ | -104.85 |
| push-back-v2–noise0.5 | $0.02 \pm 0.06$ | $\mathbf{0.04 \pm 0.05}$ | $-0.03 \pm 0.05$ | $-124.86 \pm 7.45$ | -134.63 |
| push-back-v2–noise1 | $0.0 \pm 0.01$ | $\mathbf{0.03 \pm 0.04}$ | $-0.0 \pm 0.02$ | $-143.26 \pm 6.31$ | -143.63 |
| reach-v2–noise0.1 | $\mathbf{-0.03 \pm 0.03}$ | $-0.06 \pm 0.03$ | $-0.04 \pm 0.07$ | $-17.37 \pm 0.87$ | -18.31 |
| reach-v2–noise0.5 | $\mathbf{0.05 \pm 0.13}$ | $-0.02 \pm 0.03$ | $-0.02 \pm 0.24$ | $-31.51 \pm 9.39$ | -24.57 |
| reach-v2–noise1 | $-0.12 \pm 0.42$ | $\mathbf{0.05 \pm 0.09}$ | $0.01 \pm 0.37$ | $-42.23 \pm 7.4$ | -43.79 |
| Average Relative Improvement | $-0.02$ | 0.02 | $-0.01$ | - | - |
| | Sigmoid | Rank | Constant | TD3+BC | BC |
| button-press-v2–noise0.1 | $\mathbf{0.25 \pm 0.0}$ | $0.23 \pm 0.0$ | $0.23 \pm 0.02$ | $-77.68 \pm 22.14$ | -58.79 |
| button-press-v2–noise0.5 | $\mathbf{-0.04 \pm 0.05}$ | $-0.04 \pm 0.06$ | $-0.09 \pm 0.07$ | $-59.54 \pm 1.68$ | -59.58 |
| button-press-v2–noise1 | $0.02 \pm 0.07$ | $\mathbf{0.12 \pm 0.23}$ | $0.1 \pm 0.12$ | $-81.03 \pm 5.79$ | -61.24 |
| push-back-v2–noise0.1 | $0.08 \pm 0.01$ | $-0.01 \pm 0.02$ | $\mathbf{0.05 \pm 0.04}$ | $-85.44 \pm 5.05$ | -104.85 |
| push-back-v2–noise0.5 | $\mathbf{0.07 \pm 0.07}$ | $0.05 \pm 0.14$ | $0.05 \pm 0.1$ | $-127.95 \pm 9.82$ | -134.63 |
| push-back-v2–noise1 | $\mathbf{0.01 \pm 0.04}$ | $0.01 \pm 0.06$ | $-0.04 \pm 0.02$ | $-139.72 \pm 15.6$ | -143.63 |
| reach-v2–noise0.1 | $-0.06 \pm 0.04$ | $-0.02 \pm 0.02$ | $\mathbf{-0.0 \pm 0.02}$ | $-17.11 \pm 1.28$ | -18.31 |
| reach-v2–noise0.5 | $0.12 \pm 0.19$ | $\mathbf{0.25 \pm 0.14}$ | $0.15 \pm 0.18$ | $-40.69 \pm 9.66$ | -24.57 |
| reach-v2–noise1 | $\mathbf{0.13 \pm 0.18}$ | $0.04 \pm 0.12$ | $0.1 \pm 0.11$ | $-65.48 \pm 2.85$ | -43.79 |
| Average Relative Improvement | 0.07 | 0.07 | 0.06 | - | - |

Table 10: Relative normalized score improvement for HUBL and normalized scores of base methods on MW datasets (part II)

| | Sigmoid | Rank | Constant | ATAC | BC |
|---|---|---|---|---|---|
| assembly-v2–noise0.1 | $-0.01 \pm 0.03$ | $\mathbf{0.01 \pm 0.04}$ | $-0.0 \pm 0.08$ | $-70.22 \pm 4.61$ | -71.76 |
| assembly-v2–noise0.5 | $0.01 \pm 0.02$ | $0.03 \pm 0.02$ | $\mathbf{0.08 \pm 0.0}$ | $-135.77 \pm 1.58$ | -129.53 |
| assembly-v2–noise1 | $-0.01 \pm 0.01$ | $0.02 \pm 0.01$ | $\mathbf{0.04 \pm 0.01}$ | $-140.12 \pm 2.45$ | -138.63 |
| handle-press-side-v2–noise0.1 | $-0.03 \pm 0.01$ | $\mathbf{-0.02 \pm 0.02}$ | $-0.03 \pm 0.03$ | $-32.38 \pm 0.64$ | -36.28 |
| handle-press-side-v2–noise0.5 | $0.0 \pm 0.03$ | $\mathbf{0.02 \pm 0.03}$ | $-0.01 \pm 0.03$ | $-33.04 \pm 0.96$ | -34.3 |
| handle-press-side-v2–noise1 | $\mathbf{0.04 \pm 0.12}$ | $-0.02 \pm 0.1$ | $-0.09 \pm 0.06$ | $-35.33 \pm 3.7$ | -35.72 |
| plate-slide-back-side-v2–noise0.1 | $-0.0 \pm 0.04$ | $-0.03 \pm 0.02$ | $\mathbf{0.02 \pm 0.05}$ | $-37.07 \pm 0.87$ | -37.75 |
| plate-slide-back-side-v2–noise0.5 | $0.07 \pm 0.41$ | $\mathbf{0.17 \pm 0.29}$ | $0.15 \pm 0.02$ | $-68.9 \pm 10.92$ | -65.47 |
| plate-slide-back-side-v2–noise1 | $0.17 \pm 0.3$ | $0.43 \pm 0.02$ | $\mathbf{0.44 \pm 0.11}$ | $-97.9 \pm 25.46$ | -128.28 |
| Average Relative Improvement | 0.03 | 0.07 | 0.07 | - | - |
| | Sigmoid | Rank | Constant | CQL | BC |
| assembly-v2–noise0.1 | $0.03 \pm 0.06$ | $0.06 \pm 0.17$ | $\mathbf{0.06 \pm 0.07}$ | $-76.29 \pm 9.41$ | -71.76 |
| assembly-v2–noise0.5 | $-0.0 \pm 0.02$ | $\mathbf{0.01 \pm 0.04}$ | $-0.01 \pm 0.02$ | $-131.35 \pm 2.62$ | -129.53 |
| assembly-v2–noise1 | $-0.0 \pm 0.03$ | $-0.0 \pm 0.01$ | $\mathbf{0.0 \pm 0.02}$ | $-138.94 \pm 0.97$ | -138.63 |
| handle-press-side-v2–noise0.1 | $\mathbf{0.11 \pm 0.07}$ | $-0.02 \pm 0.07$ | $0.02 \pm 0.19$ | $-34.47 \pm 4.54$ | -36.28 |
| handle-press-side-v2–noise0.5 | $0.32 \pm 0.11$ | $0.38 \pm 0.11$ | $\mathbf{0.46 \pm 0.01}$ | $-70.96 \pm 29.39$ | -34.3 |
| handle-press-side-v2–noise1 | $0.15 \pm 0.12$ | $0.13 \pm 0.2$ | $\mathbf{0.4 \pm 0.05}$ | $-55.3 \pm 33.42$ | -35.72 |
| plate-slide-back-side-v2–noise0.1 | $\mathbf{0.05 \pm 0.03}$ | $-0.0 \pm 0.11$ | $0.03 \pm 0.11$ | $-36.31 \pm 2.33$ | -37.75 |
| plate-slide-back-side-v2–noise0.5 | $\mathbf{0.03 \pm 0.09}$ | $-0.09 \pm 0.47$ | $-0.13 \pm 0.5$ | $-36.72 \pm 2.56$ | -65.47 |
| plate-slide-back-side-v2–noise1 | $0.36 \pm 0.31$ | $\mathbf{0.37 \pm 0.12}$ | $0.34 \pm 0.18$ | $-67.63 \pm 20.56$ | -128.28 |
| Average Relative Improvement | 0.12 | 0.09 | 0.13 | - | - |
| | Sigmoid | Rank | Constant | IQL | BC |
| assembly-v2–noise0.1 | $\mathbf{0.02 \pm 0.07}$ | $-0.01 \pm 0.04$ | $-0.0 \pm 0.04$ | $-72.06 \pm 2.34$ | -71.76 |
| assembly-v2–noise0.5 | $-0.01 \pm 0.01$ | $-0.01 \pm 0.02$ | $\mathbf{0.01 \pm 0.04}$ | $-122.97 \pm 6.12$ | -129.53 |
| ssembly-v2–noise1 | $-0.01 \pm 0.0$ | $-0.01 \pm 0.03$ | $\mathbf{0.0 \pm 0.03}$ | $-129.01 \pm 1.62$ | -138.63 |
| handle-press-side-v2–noise0.1 | $\mathbf{-0.05 \pm 0.01}$ | $-0.05 \pm 0.01$ | $-0.05 \pm 0.02$ | $-34.1 \pm 1.55$ | -36.28 |
| handle-press-side-v2–noise0.5 | $0.06 \pm 0.06$ | $0.05 \pm 0.07$ | $\mathbf{0.07 \pm 0.04}$ | $-37.28 \pm 0.25$ | -34.3 |
| handle-press-side-v2–noise1 | $-0.0 \pm 0.11$ | $\mathbf{0.03 \pm 0.03}$ | $0.02 \pm 0.07$ | $-46.31 \pm 3.11$ | -35.72 |
| plate-slide-back-side-v2–noise0.1 | $\mathbf{0.01 \pm 0.02}$ | $-0.01 \pm 0.01$ | $0.01 \pm 0.03$ | $-34.4 \pm 1.23$ | -37.75 |
| plate-slide-back-side-v2–noise0.5 | $\mathbf{0.07 \pm 0.1}$ | $0.04 \pm 0.08$ | $0.03 \pm 0.08$ | $-34.03 \pm 1.83$ | -65.47 |
| plate-slide-back-side-v2–noise1 | $\mathbf{0.05 \pm 0.18}$ | $0.01 \pm 0.29$ | $-0.04 \pm 0.35$ | $-43.95 \pm 10.29$ | -128.28 |
| Average Relative Improvement | 0.01 | 0.01 | 0.01 | - | - |
| | Sigmoid | Rank | Constant | TD3+BC | BC |
| assembly-v2–noise0.1 | $0.01 \pm 0.02$ | $\mathbf{0.02 \pm 0.03}$ | $-0.01 \pm 0.03$ | $-72.44 \pm 6.03$ | -71.76 |
| assembly-v2–noise0.5 | $0.0 \pm 0.02$ | $\mathbf{0.02 \pm 0.02}$ | $0.02 \pm 0.04$ | $-131.62 \pm 5.25$ | -129.53 |
| assembly-v2–noise1 | $0.0 \pm 0.0$ | $\mathbf{0.01 \pm 0.01}$ | $0.0 \pm 0.01$ | $-138.0 \pm 1.0$ | -138.63 |
| handle-press-side-v2–noise0.1 | $-0.07 \pm 0.02$ | $-0.05 \pm 0.02$ | $\mathbf{-0.05 \pm 0.02}$ | $-35.0 \pm 2.14$ | -36.28 |
| handle-press-side-v2–noise0.5 | $0.08 \pm 0.04$ | $\mathbf{0.08 \pm 0.03}$ | $0.07 \pm 0.05$ | $-40.24 \pm 1.06$ | -34.3 |
| handle-press-side-v2–noise1 | $0.08 \pm 0.08$ | $0.06 \pm 0.08$ | $\mathbf{0.13 \pm 0.17}$ | $-52.22 \pm 5.29$ | -35.72 |
| plate-slide-back-side-v2–noise0.1 | $\mathbf{0.16 \pm 0.06}$ | $0.09 \pm 0.05$ | $0.03 \pm 0.05$ | $-39.31 \pm 1.52$ | -37.75 |
| plate-slide-back-side-v2–noise0.5 | $\mathbf{0.17 \pm 0.24}$ | $-0.1 \pm 0.33$ | $-0.07 \pm 0.32$ | $-40.17 \pm 1.64$ | -65.47 |
| plate-slide-back-side-v2–noise1 | $0.19 \pm 0.35$ | $\mathbf{0.35 \pm 0.28}$ | $0.07 \pm 0.24$ | $-91.74 \pm 31.24$ | -128.28 |
| Average Relative Improvement | 0.07 | 0.05 | 0.02 | - | - |

## D.6 EXTENDED EXPERIMENTS

In this section, we study the performance of HUBL in various of scenarios.

### D.6.1 HUBL ON EXPERT DATA

We study the performance of HUBL on expert collected data. Specifically, we conduct experiments of HUBL with TD3+BC on expert datasets in D4RL. The results are reported in Table 11. Note that HUBL is able to provide performance improvements in most of the cases. But the room for improvement on such datasets is small – for any offline learning technique. Therefore, due to finite sample error, HUBL may occasionally hurt performance empirically

|  | Constant | Rank | Sigmoid | Base Performance |
|---|---|---|---|---|
| halfcheetah-expert-v2 | 1% | 0% | 1% | 97.09 |
| hopper-expert-v2 | 3% | 3% | 3% | 108.46 |
| walker2d-expert-v2 | 0% | 1% | 0% | 110.14 |

Table 11: Relative improvement of HUBL for TD3+BC on expert data

### D.6.2 HUBL WITH ONLINE RL METHODS

We provide experiments where we combine HUBL with DDPG on D4RL datasets, without any additional designs for the offline RL setting. The results are provided in Table 12.

| Performance of DDPG | 13.43 |
|---|---|
| Relative improvement of HUBL with sigmoid | 53% |
| Relative improvement of HUBL with constant | 74% |
| Relative improvement of HUBL with portion | 90% |

Table 12: Relative improvement of HUBL for DDPG on D4RL datasets

Note that HUBL is able to significantly improve DDPG's performance in relative terms. However, the absolute performance of DDPG+HuBL is much worse than other methods we considered in the main paper. This is expected, as HUBL is designed to augment offline RL algorithms, as opposed to making a general online RL algorithm (like DDPG) work also offline.

To see this more clearly, we consider a toy example with three actions $a_1$, $a_2$, $a_3$ at state $s_0$. Suppose that and $a_1$ and $a_2$ are all out-of-support. As a result, without additional designs (such as pessimism) for offline RL, the policy may take $a_1$ and $a_2$ regardless how we set the heuristic, since the heuristic here would only affect the value for in-support actions and not the value for out-of-support actions. Thus, HUBL cannot turn an online RL algorithm to an offline RL algorithm; it can only enhance an algorithm that is designed for offline RL already.

### D.6.3 HUBL IN STOCHASTIC ENVIRONMENTS

To test the performance of HUBL in stochastic environments, we construct a random version of walker2d-v2 by adding $\beta\epsilon$ to the action passed to the environment, where $\epsilon$ is a uniform random variable taking values in $[-1, 1]$, and $\beta$ is a scalar controlling the amount of randomness. Note that we keep the scale of noise in $[-1, 1]$, and the action variable itself is normalized to $[-1, 1]$ in D4RL. With the new environment, we regenerate the data using the original walker2d-meidum-v2 policy and apply TD3+BC with HUBL. The results are presented in Table 13.

|  | Sigmoid | Rank | Constant | Base Performance |
|---|---|---|---|---|
| $\beta = 0.0001$ | 6% | 6% | 4% | 66.99 |
| $\beta = 0.001$ | 6% | 8% | 7% | 66.73 |
| $\beta = 0.01$ | 4% | 4% | 6% | 64.23 |
| $\beta = 1$ | 4% | -4% | 5% | 28.09 |

Table 13: Relative improvement of HUBL for TD3+BC on stochastic walker2d-meidum-v2

The base performance decreases as the noise increases, as expected. However, HUBL is still able to provide some performance improvement under this stochastic setting. We also notice that the constant labeling function is more robust to the noise. This makes sense as the constant function is not sensitive to the heuristic estimation. But if there is more noise, the performance of HUBL will further degenerate, which is a limitation of HUBL.

## D.7 Extended Results for Absolute Improvement

We report the experiment results in absolute improvement of normalized score for D4RL and Meta-World.

Table 14: Normalized score improvement for HUBL and normalized scores of base methods on D4RL datasets.

| | Sigmoid | Rank | Constant | ATAC | BC |
|---|---|---|---|---|---|
| halfcheetah-medium-expert-v2 | **2.13 ± 0.35** | 1.55 ± 0.96 | 1.76 ± 0.14 | 92.5 ± 0.73 | 45.83 |
| halfcheetah-medium-replay-v2 | 0.43 ± 0.38 | 0.96 ± 0.27 | **1.04 ± 1.01** | 46.89 ± 0.25 | 37.73 |
| halfcheetah-medium-v2 | −0.05 ± 0.71 | 0.16 ± 0.25 | **0.85 ± 0.35** | 52.54 ± 0.63 | 42.92 |
| hopper-medium-expert-v2 | 0.36 ± 0.03 | **0.44 ± 0.26** | 0.27 ± 0.1 | 111.04 ± 0.13 | 57.51 |
| hopper-medium-replay-v2 | 0.01 ± 0.48 | 0.21 ± 0.77 | **0.26 ± 0.32** | 101.23 ± 0.59 | 32.22 |
| hopper-medium-v2 | 0.44 ± 3.28 | **1.57 ± 6.09** | −2.35 ± 12.85 | 85.59 ± 4.76 | 53.81 |
| walker2d-medium-expert-v2 | 0.08 ± 0.4 | **1.72 ± 0.5** | 0.26 ± 0.6 | 111.99 ± 0.8 | 105.3 |
| walker2d-medium-replay-v2 | 2.76 ± 2.23 | **7.15 ± 0.31** | 4.38 ± 6.53 | 84.22 ± 2.7 | 22.78 |
| walker2d-medium-v2 | 0.6 ± 0.5 | −1.95 ± 4.62 | **1.35 ± 0.17** | 87.2 ± 0.67 | 64.18 |
| Average Improvement | 0.75 | 1.31 | 0.87 | - | - |
| | Sigmoid | Rank | Constant | CQL | BC |
| halfcheetah-medium-expert-v2 | 1.55 ± 1.75 | **2.66 ± 4.77** | −1.49 ± 13.59 | 81.96 ± 9.93 | 45.83 |
| halfcheetah-medium-replay-v2 | 4.14 ± 0.22 | **7.19 ± 0.73** | 5.26 ± 0.4 | 40.98 ± 0.81 | 37.73 |
| halfcheetah-medium-v2 | −5.12 ± 0.43 | **−0.86 ± 0.72** | −1.35 ± 0.68 | 51.05 ± 0.83 | 42.92 |
| hopper-medium-expert-v2 | **32.62 ± 1.08** | 30.72 ± 5.21 | 29.5 ± 3.76 | 78.39 ± 14.51 | 57.51 |
| hopper-medium-replay-v2 | **3.29 ± 0.79** | 0.69 ± 1.31 | 2.09 ± 0.75 | 97.21 ± 3.15 | 32.22 |
| hopper-medium-v2 | **12.81 ± 10.16** | 10.87 ± 11.68 | 12.01 ± 4.51 | 70.27 ± 8.77 | 53.81 |
| walker2d-medium-expert-v2 | 19.14 ± 0.42 | **19.74 ± 0.82** | 19.36 ± 0.14 | 89.38 ± 17.54 | 105.3 |
| walker2d-medium-replay-v2 | 4.84 ± 17.24 | **24.07 ± 3.58** | 23.4 ± 2.24 | 63.35 ± 6.22 | 22.78 |
| walker2d-medium-v2 | 3.62 ± 3.86 | **4.14 ± 2.23** | −2.01 ± 11.8 | 79.41 ± 1.82 | 64.18 |
| Average Improvement | 8.54 | 11.02 | 9.64 | - | - |
| | Sigmoid | Rank | Constant | IQL | BC |
| halfcheetah-medium-expert-v2 | **0.2 ± 0.9** | −0.28 ± 0.19 | −4.59 ± 2.6 | 92.92 ± 0.5 | 45.83 |
| halfcheetah-medium-replay-v2 | −1.51 ± 0.67 | **−0.08 ± 0.83** | −1.86 ± 1.21 | 44.35 ± 0.21 | 37.73 |
| halfcheetah-medium-v2 | −0.28 ± 0.13 | **−0.06 ± 0.23** | −0.68 ± 0.36 | 49.61 ± 0.16 | 42.92 |
| hopper-medium-expert-v2 | −4.52 ± 3.29 | **−2.88 ± 2.08** | −10.22 ± 12.89 | 108.13 ± 2.69 | 57.51 |
| hopper-medium-replay-v2 | 38.66 ± 4.79 | 31.57 ± 13.41 | 33.41 ± 14.48 | **53.98 ± 4.18** | 32.22 |
| hopper-medium-v2 | **1.78 ± 2.05** | 1.27 ± 1.69 | −0.68 ± 8.99 | 61.03 ± 3.27 | 53.81 |
| walker2d-medium-expert-v2 | **1.53 ± 0.31** | 1.06 ± 1.01 | 0.99 ± 0.28 | 109.59 ± 0.2 | 105.3 |
| walker2d-medium-replay-v2 | 10.78 ± 9.11 | **15.49 ± 1.61** | 9.2 ± 3.04 | 73.57 ± 2.64 | 22.78 |
| walker2d-medium-v2 | 2.17 ± 1.46 | **2.54 ± 0.9** | 3.0 ± 0.71 | 81.68 ± 2.48 | 64.18 |
| Average Improvement | 5.42 | 5.4 | 3.17 | - | - |
| | Sigmoid | Rank | Constant | TD3+BC | BC |
| halfcheetah-medium-expert-v2 | **−1.52 ± 2.92** | −2.01 ± 2.51 | −1.74 ± 2.29 | 94.82 ± 0.33 | 45.83 |
| halfcheetah-medium-replay-v2 | −5.48 ± 1.47 | −4.41 ± 8.83 | **−0.24 ± 0.82** | 54.38 ± 1.24 | 37.73 |
| halfcheetah-medium-v2 | −4.28 ± 1.5 | −2.85 ± 0.84 | **−0.76 ± 0.94** | 64.63 ± 2.15 | 42.92 |
| hopper-medium-expert-v2 | −5.8 ± 13.25 | −15.45 ± 19.55 | **2.61 ± 4.79** | 98.79 ± 12.57 | 57.51 |
| hopper-medium-replay-v2 | 13.36 ± 6.4 | 17.23 ± 2.15 | **19.7 ± 2.13** | 81.43 ± 27.77 | 32.22 |
| hopper-medium-v2 | 30.06 ± 10.23 | **41.58 ± 1.11** | 35.2 ± 11.11 | 59.12 ± 3.39 | 53.81 |
| walker2d-medium-expert-v2 | 0.36 ± 0.84 | **0.92 ± 0.95** | 0.53 ± 0.35 | 111.87 ± 0.17 | 105.3 |
| walker2d-medium-replay-v2 | **9.85 ± 3.1** | 7.83 ± 5.66 | 3.86 ± 1.91 | 84.95 ± 1.85 | 22.78 |
| walker2d-medium-v2 | −0.75 ± 2.86 | **6.81 ± 1.13** | 0.37 ± 5.76 | 84.09 ± 1.94 | 64.18 |
| Average Improvement | 4.23 | 5.76 | 6.86 | - | - |

Table 15: Normalized score improvement for HUBL and normalized scores of base methods on MW datasets (part I)

| | Sigmoid | Rank | Constant | ATAC | BC |
|---|---|---|---|---|---|
| button-press-v2–noise0.1 | $-0.09 \pm 1.76$ | $\mathbf{0.43 \pm 0.62}$ | $0.41 \pm 0.82$ | $-58.12 \pm 0.45$ | -58.79 |
| button-press-v2–noise0.5 | $-0.21 \pm 1.8$ | $-2.22 \pm 3.5$ | $\mathbf{0.86 \pm 2.75}$ | $-64.16 \pm 0.93$ | -59.58 |
| button-press-v2–noise1 | $-3.42 \pm 4.41$ | $\mathbf{-0.23 \pm 2.71}$ | $-0.92 \pm 2.28$ | $-62.72 \pm 3.84$ | -61.24 |
| push-back-v2–noise0.1 | $8.81 \pm 2.64$ | $3.59 \pm 6.53$ | $\mathbf{12.75 \pm 5.12}$ | $-87.12 \pm 13.89$ | -104.85 |
| push-back-v2–noise0.5 | $14.47 \pm 8.88$ | $\mathbf{25.88 \pm 6.55}$ | $22.0 \pm 3.23$ | $-133.28 \pm 14.24$ | -134.63 |
| push-back-v2–noise1 | $5.38 \pm 7.51$ | $3.28 \pm 12.83$ | $\mathbf{7.94 \pm 9.56}$ | $-141.29 \pm 13.87$ | -143.63 |
| reach-v2–noise0.1 | $-0.03 \pm 1.08$ | $-0.04 \pm 0.9$ | $\mathbf{0.0 \pm 1.14}$ | $-17.19 \pm 0.76$ | -18.31 |
| reach-v2–noise0.5 | $12.53 \pm 2.27$ | $\mathbf{15.6 \pm 1.9}$ | $14.49 \pm 2.68$ | $-33.91 \pm 5.62$ | -24.57 |
| reach-v2–noise1 | $\mathbf{18.22 \pm 2.13}$ | $16.14 \pm 11.32$ | $23.09 \pm 2.54$ | $-46.58 \pm 3.98$ | -43.79 |
| Average Improvement | 6.18 | 6.94 | 8.96 | - | - |
| | Sigmoid | Rank | Constant | CQL | BC |
| button-press-v2–noise0.1 | $0.23 \pm 1.96$ | $1.32 \pm 2.71$ | $\mathbf{1.51 \pm 1.87}$ | $-59.67 \pm 1.69$ | -58.79 |
| button-press-v2–noise0.5 | $-2.29 \pm 2.67$ | $\mathbf{-1.18 \pm 0.55}$ | $-1.37 \pm 0.19$ | $-58.48 \pm 0.85$ | -59.58 |
| button-press-v2–noise1 | $-5.37 \pm 7.82$ | $\mathbf{5.46 \pm 1.68}$ | $2.71 \pm 7.2$ | $-67.18 \pm 14.25$ | -61.24 |
| push-back-v2–noise0.1 | $-5.54 \pm 5.79$ | $\mathbf{1.54 \pm 3.86}$ | $1.33 \pm 1.89$ | $-81.78 \pm 2.88$ | -104.85 |
| push-back-v2–noise0.5 | $\mathbf{30.82 \pm 14.6}$ | $27.34 \pm 12.97$ | $23.15 \pm 14.97$ | $-124.64 \pm 2.45$ | -134.63 |
| push-back-v2–noise1 | $7.69 \pm 16.81$ | $\mathbf{14.41 \pm 19.38}$ | $13.58 \pm 7.12$ | $-133.66 \pm 8.17$ | -143.63 |
| reach-v2–noise0.1 | $-0.77 \pm 3.1$ | $\mathbf{-0.22 \pm 0.98}$ | $-0.67 \pm 0.41$ | $-18.5 \pm 2.52$ | -18.31 |
| reach-v2–noise0.5 | $\mathbf{21.05 \pm 1.89}$ | $7.28 \pm 21.76$ | $16.49 \pm 11.16$ | $-41.02 \pm 18.79$ | -24.57 |
| reach-v2–noise1 | $86.19 \pm 2.49$ | $37.15 \pm 15.77$ | $\mathbf{86.65 \pm 3.53}$ | $-107.14 \pm 32.75$ | -43.79 |
| Average Improvement | 14.67 | 10.34 | 15.93 | - | - |
| | Sigmoid | Rank | Constant | IQL | bc |
| button-press-v2–noise0.1 | $0.53 \pm 0.6$ | $\mathbf{1.87 \pm 0.75}$ | $1.83 \pm 0.58$ | $-59.87 \pm 1.32$ | -58.79 |
| button-press-v2–noise0.5 | $-1.09 \pm 1.25$ | $\mathbf{-0.63 \pm 1.16}$ | $-8.45 \pm 7.68$ | $-60.68 \pm 0.29$ | -59.58 |
| button-press-v2–noise1 | $-1.61 \pm 8.16$ | $\mathbf{5.17 \pm 0.72}$ | $3.19 \pm 2.58$ | $-73.96 \pm 5.09$ | -61.24 |
| push-back-v2–noise0.1 | $2.93 \pm 3.28$ | $\mathbf{3.11 \pm 1.89}$ | $2.47 \pm 0.59$ | $-83.79 \pm 5.45$ | -104.85 |
| push-back-v2–noise0.5 | $0.39 \pm 11.3$ | $\mathbf{4.62 \pm 6.24}$ | $-4.16 \pm 6.09$ | $-124.86 \pm 7.45$ | -134.63 |
| push-back-v2–noise1 | $1.82 \pm 5.35$ | $\mathbf{3.71 \pm 5.69}$ | $-0.35 \pm 3.1$ | $-143.26 \pm 6.31$ | -143.63 |
| reach-v2–noise0.1 | $\mathbf{-0.55 \pm 0.2}$ | $-1.0 \pm 0.45$ | $-0.68 \pm 1.23$ | $-17.37 \pm 0.87$ | -18.31 |
| reach-v2–noise0.5 | $-2.48 \pm 4.95$ | $\mathbf{-0.58 \pm 0.95}$ | $-0.58 \pm 7.66$ | $-31.51 \pm 9.39$ | -24.57 |
| reach-v2–noise1 | $-3.69 \pm 5.46$ | $\mathbf{2.12 \pm 4.0}$ | $0.33 \pm 15.48$ | $-42.23 \pm 7.4$ | -43.79 |
| Average Relative Improvement | $-0.41$ | 2.04 | $-0.71$ | - | - |
| | Sigmoid | Rank | Constant | TD3+BC | BC |
| button-press-v2–noise0.1 | $\mathbf{19.12 \pm 0.22}$ | $18.19 \pm 0.17$ | $18.01 \pm 1.73$ | $-77.68 \pm 22.14$ | -58.79 |
| button-press-v2–noise0.5 | $-2.47 \pm 2.73$ | $\mathbf{-2.11 \pm 3.75}$ | $-5.36 \pm 4.18$ | $-59.54 \pm 1.68$ | -59.58 |
| button-press-v2–noise1 | $1.71 \pm 5.97$ | $\mathbf{9.71 \pm 18.76}$ | $8.4 \pm 9.47$ | $-81.03 \pm 5.79$ | -61.24 |
| push-back-v2–noise0.1 | $\mathbf{7.18 \pm 0.66}$ | $-1.06 \pm 1.65$ | $4.11 \pm 3.08$ | $-85.44 \pm 5.05$ | -104.85 |
| push-back-v2–noise0.5 | $\mathbf{9.43 \pm 8.7}$ | $6.1 \pm 17.6$ | $6.43 \pm 12.98$ | $-127.95 \pm 9.82$ | -134.63 |
| push-back-v2–noise1 | $\mathbf{1.26 \pm 6.0}$ | $1.4 \pm 8.05$ | $-5.31 \pm 2.35$ | $-139.72 \pm 15.6$ | -143.63 |
| reach-v2–noise0.1 | $-0.99 \pm 0.66$ | $-0.28 \pm 0.31$ | $\mathbf{-0.07 \pm 0.38}$ | $-17.11 \pm 1.28$ | -18.31 |
| reach-v2–noise0.5 | $5.04 \pm 7.92$ | $\mathbf{10.2 \pm 5.67}$ | $6.07 \pm 7.23$ | $-40.69 \pm 9.66$ | -24.57 |
| reach-v2–noise1 | $\mathbf{8.4 \pm 12.06}$ | $2.34 \pm 7.98$ | $6.74 \pm 7.36$ | $-65.48 \pm 2.85$ | -43.79 |
| Average Improvement | 5.41 | 4.94 | 4.34 | - | - |

Table 16: Normalized score improvement for HUBL and normalized scores of base methods on MW datasets (part II)

| | Sigmoid | Rank | Constant | ATAC | BC |
|---|---|---|---|---|---|
| assembly-v2–noise0.1 | −0.99 ± 2.28 | **1.0 ± 2.73** | −0.23 ± 5.65 | −70.22 ± 4.61 | -71.76 |
| assembly-v2–noise0.5 | 1.05 ± 2.43 | 3.59 ± 3.02 | **11.13 ± 6.31** | −135.77 ± 1.58 | -129.53 |
| assembly-v2–noise1 | −1.29 ± 1.99 | 2.47 ± 1.95 | **5.75 ± 1.07** | −140.12 ± 2.45 | -138.63 |
| handle-press-side-v2–noise0.1 | −0.93 ± 0.41 | **−0.49 ± 0.64** | −0.88 ± 1.06 | −32.38 ± 0.64 | -36.28 |
| handle-press-side-v2–noise0.5 | 0.08 ± 0.86 | **0.74 ± 0.94** | −0.35 ± 1.06 | −33.04 ± 0.96 | -34.3 |
| handle-press-side-v2–noise1 | **1.25 ± 4.2** | −0.73 ± 3.55 | −3.07 ± 2.05 | −35.33 ± 3.7 | -35.72 |
| plate-slide-back-side-v2–noise0.1 | −0.07 ± 1.44 | −0.98 ± 0.82 | **0.77 ± 1.96** | −37.07 ± 0.87 | -37.75 |
| plate-slide-back-side-v2–noise0.5 | 5.15 ± 28.23 | **11.72 ± 20.14** | 10.47 ± 1.41 | −68.9 ± 10.92 | -65.47 |
| plate-slide-back-side-v2–noise1 | 16.4 ± 29.75 | 42.45 ± 2.22 | **43.25 ± 10.94** | −97.9 ± 25.46 | -128.28 |
| Average Relative Improvement | 2.29 | 6.64 | 7.43 | - | - |
| | Sigmoid | Rank | Constant | CQL | BC |
| assembly-v2–noise0.1 | 2.37 ± 4.62 | 4.2 ± 12.62 | **4.36 ± 5.55** | −76.29 ± 9.41 | -71.76 |
| assembly-v2–noise0.5 | −0.57 ± 2.09 | **1.44 ± 5.55** | −1.21 ± 2.87 | −131.35 ± 2.62 | -129.53 |
| assembly-v2–noise1 | −0.01 ± 3.97 | −0.35 ± 2.07 | **0.56 ± 2.56** | −138.94 ± 0.97 | -138.63 |
| handle-press-side-v2–noise0.1 | **3.79 ± 2.52** | −0.53 ± 2.53 | 0.73 ± 6.56 | −34.47 ± 4.54 | -36.28 |
| handle-press-side-v2–noise0.5 | 23.05 ± 7.99 | 26.8 ± 8.13 | **32.31 ± 0.94** | −70.96 ± 29.39 | -34.3 |
| handle-press-side-v2–noise1 | 8.33 ± 6.37 | 6.94 ± 10.92 | **22.01 ± 2.89** | −55.3 ± 33.42 | -35.72 |
| plate-slide-back-side-v2–noise0.1 | **1.83 ± 1.21** | −0.02 ± 3.82 | 0.95 ± 3.95 | −36.31 ± 2.33 | -37.75 |
| plate-slide-back-side-v2–noise0.5 | **0.96 ± 3.25** | −3.16 ± 17.37 | −4.94 ± 18.35 | −36.72 ± 2.56 | -65.47 |
| plate-slide-back-side-v2–noise1 | 24.46 ± 21.13 | **24.72 ± 8.42** | 23.04 ± 12.37 | −67.63 ± 20.56 | -128.28 |
| Average Relative Improvement | 7.14 | 6.67 | 8.65 | - | - |
| | Sigmoid | Rank | Constant | IQL | BC |
| assembly-v2–noise0.1 | **1.18 ± 4.78** | −0.49 ± 3.07 | −0.02 ± 3.2 | −72.06 ± 2.34 | -71.76 |
| assembly-v2–noise0.5 | −1.49 ± 1.27 | −0.9 ± 1.89 | **1.13 ± 4.46** | −122.97 ± 6.12 | -129.53 |
| assembly-v2–noise1 | −1.31 ± 0.33 | −0.76 ± 3.55 | **0.39 ± 3.85** | −129.01 ± 1.62 | -138.63 |
| handle-press-side-v2–noise0.1 | **−1.56 ± 0.5** | −1.67 ± 0.39 | −1.66 ± 0.53 | −34.1 ± 1.55 | -36.28 |
| handle-press-side-v2–noise0.5 | 2.16 ± 2.07 | 2.04 ± 2.46 | **2.56 ± 1.54** | −37.28 ± 0.25 | -34.3 |
| handle-press-side-v2–noise1 | −0.2 ± 5.06 | **1.33 ± 1.52** | 1.12 ± 3.27 | −46.31 ± 3.11 | -35.72 |
| plate-slide-back-side-v2–noise0.1 | 0.24 ± 0.76 | −0.18 ± 0.48 | **0.42 ± 1.12** | −34.4 ± 1.23 | -37.75 |
| plate-slide-back-side-v2–noise0.5 | **2.3 ± 3.47** | 1.3 ± 2.69 | 0.92 ± 2.7 | −34.03 ± 1.83 | -65.47 |
| plate-slide-back-side-v2–noise1 | **2.33 ± 7.7** | 0.63 ± 12.54 | −1.8 ± 15.22 | −43.95 ± 10.29 | -128.28 |
| Average Relative Improvement | 0.41 | 0.14 | 0.34 | - | - |
| | Sigmoid | Rank | Constant | TD3+BC | BC |
| assembly-v2–noise0.1 | 1.08 ± 1.69 | **1.16 ± 2.35** | −0.71 ± 1.95 | −72.44 ± 6.03 | -71.76 |
| assembly-v2–noise0.5 | 0.56 ± 3.01 | 2.16 ± 3.05 | **2.6 ± 5.8** | −131.62 ± 5.25 | -129.53 |
| assembly-v2–noise1 | 0.48 ± 0.6 | **0.71 ± 1.39** | 0.03 ± 1.92 | −138.0 ± 1.0 | -138.63 |
| handle-press-side-v2–noise0.1 | −2.34 ± 0.77 | −1.91 ± 0.87 | **−1.67 ± 0.69** | −35.0 ± 2.14 | -36.28 |
| handle-press-side-v2–noise0.5 | **3.25 ± 1.66** | 3.17 ± 1.04 | 2.97 ± 1.82 | −40.24 ± 1.06 | -34.3 |
| handle-press-side-v2–noise1 | 4.31 ± 4.29 | 3.39 ± 3.97 | **6.93 ± 8.89** | −52.22 ± 5.29 | -35.72 |
| plate-slide-back-side-v2–noise0.1 | **6.11 ± 2.3** | 3.67 ± 1.77 | 1.09 ± 2.11 | −39.31 ± 1.52 | -37.75 |
| plate-slide-back-side-v2–noise0.5 | **6.77 ± 9.5** | −4.01 ± 13.35 | −2.91 ± 13.02 | −40.17 ± 1.64 | -65.47 |
| plate-slide-back-side-v2–noise1 | 17.06 ± 32.1 | **31.78 ± 25.94** | 6.48 ± 22.36 | −91.74 ± 31.24 | -128.28 |
| Average Relative Improvement | 4.14 | 4.46 | 1.65 | - | - |

# E    LIMITATIONS AND FUTURE WORK

In our current framework, HUBL calculates heuristic values as Monte-Carlo returns. This is feasible in practical scenarios where data comes in the form of trajectories, but much of offline RL literature experiments with batch datasets that consist of disconnected transition tuples. For such datasets, HUBL requires other heuristic calculation strategies.

Empirically, we have showed HUBL's utility on benchmarks where observations are full-information low-dimensional state vectors and state transitions are deterministic. While these benchmarks are standard in offline RL, evaluation in more stochastic domains where observations are partial and high-dimensional, such as robotics, will give a more complete picture of HUBL's utility. So would evaluating HUBL with model-based offline RL such as MOReL (Kidambi et al., 2020). Also, HUBL is not applicable to bootstrapping-free offline RL methods (Schaefer et al., 2007; Deisenroth and Rasmussen, 2011; Depeweg et al., 2016; Swazinna et al., 2021; 2022).

On the theory side, our analysis focuses on state-dependent heuristics. In practice, though, heuristic estimates commonly depend on full trajectories, and our theory needs adjustments to account for this. Same goes for applying our theory to model-based offline RL such as MOReL.

For future direction, we aim to develop better heuristics and blending strategies for batch datasets and sparse-reward problems. We also plan to apply HUBL to other tasks in healthcare and finance (Alaluf et al., 2022; Geng et al., 2018; 2019a;b) where collecting data is especially expensive.

