# OpenReview forum: "Improving Offline RL by Blending Heuristics"
_ICLR.cc/2024/Conference — ICLR 2024 spotlight_

### Official Review · Reviewer_iqAo · 2023-10-25

**Soundness:** 4 excellent
**Presentation:** 4 excellent
**Contribution:** 3 good
**Rating:** 8
**Confidence:** 4

**Summary:**

The authors propose a technique known as Heuristic Blending (HUBL) to enhance the performance of model-free offline Reinforcement Learning (RL) algorithms that are based on value bootstrapping. The primary aim of this approach is to mitigate the challenges associated with bootstrapping and achieve stable performance.

HUBL essentially adapts the rewards and discount factor within the offline dataset utilized by the base offline RL algorithm. It achieves this by modifying the reward through a blending process, combining it with a state-specific heuristic derived from the Monte Carlo return of the behavior policy. It also reduces the discount factor discount factor. The degree of reduction in the discount factor and the reward blending is determined by a trajectory-dependent blending factor. This factor is designed to be high for trajectories in which the behavior policy performs well and low otherwise.

The authors offer three distinct methods for selecting this blending factor. They support their algorithm with theoretical analysis and experimental results.

**Strengths:**

1. The proposed algorithm is simple and can be implemented with minimal overhead.
2. The authors provide a complete theoretical analysis, and also conduct extensive experiments, on both deterministic and stochastic environments (although just one) .
3. The paper is well written and easy to follow.

**Weaknesses:**

$\tilde{r}$ **dependence on the behavior policy may cause problems when data is collected using multiple behavior policies.**

Consider a dataset with data from a mixture of policies (say  the medium replay case in D4RL), and for simplicity suppose we assume  a constant lambda, say 0.5. Now, since the reward $\tilde{r}$ is conditioned on the behavior policy, doesn’t the effective MDP (reconstructed using the relabeled dataset) become non stationary? Is the performance not affected much due to the deterministic nature of the environments chosen for evaluation?

**Choosing $\lambda$ may not be straightforward**

The choice of $\lambda$ should be such that it is high for trajectories with high returns, and low otherwise. But this is hard to determine if the dataset consists of trajectories that perform equally well. This can be seen in results shown in figure 2, as significant improvement is seen on datasets with with a mixture of behavior policies because there is a way to determine the appropriate choice of $\lambda$ based on relative performance. Now, suppose you have a dataset with low rewards (the random variant in D4RL), then the ranking method will still assign high ranks to most trajectories, and this might result in poor performance as the Monte Carlo estimates using the behavior policy might cause instability.

**Questions:**

See weaknesses, in addition to them,

1. Why is the training of a value function needed during step 2?

---

> ### Author Response · Authors · 2023-11-22
> **Thank you for your constructive comment. We have have conducted additional experiments as you suggested.**
>
> Thank you very much for your constructive feedback and inspiring comment on our draft. We hope to answer your questions by providing more theoretical details and additional experiments.
>
> > r dependence on the behavior policy may cause problems when data is collected using multiple behavior policies.
>
> In the theoretical analysis, we made a simplifying assumption that the heuristic $h$ is a function of the state in Theorem 1. In this case, HUBL can be viewed as solving a stationary reshaped MDP. In Theorem 2, we further consider a special case that  $h(s)$ is the value function of a policy e.g. the behavior policy $\mu$ when proving an exact finite sample upper bound. This extra assumption that the heuristic is a policy’s value function is made as a sufficient condition to establish the heuristic is improvable (see [Definition 4.1, Cheng et al., 2021]), which roughly means that the heuristic value is attainable and as a result can guide the learning (i.e. $\tilde{V}^*(s) \geq h(s)$ [Proposition 4.4, Cheng et al., 2021]). This sufficient condition is used to show the second term in the regret  $h(s’) - \tilde{V}^\pi(s’)$ can be non-positive:
>
> $\textnormal{Regret}(\pi,  h, \lambda):=
>          \tilde{V}^{\pi^*}(d_0) - \tilde{V}^{{\pi}}(d_0) + \frac{\gamma}{1-\gamma}E_{(s,a,s')\sim d^\pi}[\lambda(s') ( h(s') - \tilde{V}^{{\pi}}(s')) | s, s'\in\Omega ]$
>
> On the other hand, we totally agree that in practical scenarios with mixed behavior policies, the estimated heuristics in Section 4.2 may lead to a non-stationary MDP. As discussed above, the current analysis would work through if this heuristic (in expectation) turns out to be improvable. We think analyzing the exact condition for this to happen in a scenario of mixed policies is interesting, but it’s beyond the scope of the current work. When the resulting heuristic is not improvable, it eventually will lead to an extra error term in Section 4.2. As a result, the performance of HUBL is affected by how well the heuristic function is calculated, which is affected by many factors like the randomness of the environment as you pointed out, the amount of data, the coverage of the data, and so on. Specifically for the randomness in environments, we provided experiments of HUBL with stochastic environments in Section D.3, where HUBL still provides performance improvement.
>
> > Choosing lambda may not be straightforward
>
> When the trajectories perform equally well, HUBL with Rank labeling behaves exactly as you expected: the trajectories will have similar lambda. However, this does not mean that HUBL will necessarily hurt the performance, as there is $\alpha <1$ discounting the amount of heuristics blended into bootstrapping. Empirically, we applied HUBL with Rank labeling on hopper-random-v2, walker2d-random-v2 and halfcheetah-random-v2. The results are provided below. We can see that HUBL is able to improve the performance. However, in this setting, the performance improvement of HUBL is limited. We believe the reason is as you suggested: the $\lambda$ function could not find well-performing trajectories to blend with bootstrapping. A better design for $\lambda$ is required for such scenarios, which is one of our future directions. We will include such discussions in the final version   of the submission.
> |                       | Relative Performance Improvement of HUBL | Base  |
> | --------------------- | ---------------------------------------- | ----- |
> | hopper-random-v2      | 4%                                       | 15.18 |
> | walker2d-random-v2    | 7%                                       | 3.94  |
> | halfcheetah-random-v2 | 0%                                       | 26.1  |
>
> > Why is the training of a value function needed in step 2.
>
> We are not quite sure which step you are referring to as we do not train a value function in Step 2 on page 4. We train a value function in Step 1: Computing heuristic $h_t$ on page 4 (footnote 3). That is because sometimes a trajectory ends not due to reaching a terminal stage but due to a timeout. When this happens, the simple Monte-Carlo sum of the rewards underestimates the value of this trajectory. Therefore, we need to estimate a value function to add at the end of such trajectories.

---

> > ### Comment · Reviewer_iqAo · 2023-11-22
> >
> > I thank the authors for their detailed response, and the additional experiments. I have modified my score accordingly. Apologies for the confusion with my question, I was indeed referring to the footnote on page 4.

---

> > > ### Author Response · Authors · 2023-11-22
> > >
> > > Thank you very much for your reply. We are glad that our response is helpful. If you have any additional questions or new comments, please do not hesitate to let us now.

---

### Official Review · Reviewer_zyzv · 2023-10-26

**Soundness:** 3 good
**Presentation:** 4 excellent
**Contribution:** 3 good
**Rating:** 8
**Confidence:** 4

**Summary:**

The paper deals with offline RL. It focuses on Q-function based offline RL methods and presents a heuristic to increase the performance of these methods. In extensive experiments a performance increase is observed on average.

**Strengths:**

* The presentation is very good.  I would like to emphasize that the limitations mention both the restriction to trajectories and the lack of stochastic MDPs among the benchmarks used. This is exemplary. Also that already in the first sentence `We propose Heuristic Blending (HUBL), a simple performance-improving technique for a broad class of offline RL algorithms based on value bootstrapping` it is clearly stated to which class of algorithms the paper refers.
* The method is investigated as a modification of not just one, but four state of the art bootstrapping-based offline RL algorithms.

**Weaknesses:**

none

**Questions:**

No questions, but a few notes and comments:
* "methd" -> "method"

* At "Step 1: Computing heuristic ht" ht should be set boldmath, analogously in Step 2 and Step 3.

* At `Justin Fu, Aviral Kumar, Ofir Nachum, George Tucker, and Sergey Levine. D4rl: Datasets for deep
data-driven reinforcement learning, 2020` the indication where it was published is missing.

* In the references there are some unintentional lower case letters, e.g. mdp, Monte carlo.

* Just for completeness, I’d like to point out that there are also purely model-based batch/offline RL methods that do not use Q-function and are thus bootstrapping-free [1-4]. See [5] for a discussion. Since the authors have precisely formulated at the very beginning that this paper deals with the algorithm class with Q-function and bootstrapping, a mention of these bootstrapping-free algorithms is probably not necessary.

[1] Schaefer et al., A recurrent control neural network for data efficient reinforcement learning, 2007\
[2] Deisenroth and Rasmussen, PILCO: A Model-Based and Data-Efficient Approach to Policy Search, 2011\
[3] Depeweg et al., Learning and policy search in stochastic dynamical systems with Bayesian neural networks, 2017\
[4] Swazinna et al., Overcoming model bias for robust offline deep reinforcement learning, 2021\
[5] Swazinna et al., Comparing Model-free and Model-based Algorithms for Offline Reinforcement Learning, 2022

---

> ### Author Response · Authors · 2023-11-22
> **Thank you very much for your positive feedback.**
>
> Thank you very much for your positive feedback and pointing out additional references. We have updated the draft addressing the presentation issues and additional citations you mentioned.

---

### Official Review · Reviewer_Gwab · 2023-10-31

**Soundness:** 4 excellent
**Presentation:** 3 good
**Contribution:** 4 excellent
**Rating:** 8
**Confidence:** 4

**Summary:**

This work presents Heuristic Blending (HUBL), a new technique that can be attached to offline RL methods to improve their performance.

HUBL works by modifying the rewards and discounts in the dataset that the offline RL algorithm consumes, blending heuristic values to partially replace bootstrapping.

Theoretical and empirical results show HUBL consistency when improving several offline RL algorithms performance by 9% on average over D4RL and Meta World datasets.

**Strengths:**

* This paper meets very good originality, quality, clarity and significance criteria. Good job!

* Section 2 does analyze the main differences of the proposed approach with respect to cited works. It is clear that no previous model has addressed the data relabeling as it has been proposed in this manuscript for the particular setting of offline RL. The use of both data relabeling and heuristic in combination with RL has been explored before, but not in the offline scenario.

* HUBL is an original method to incorporate into existing offline RL methods. It improves them without any need for a complicated modification within any algorithm; only a relabeling of the dataset is needed. It is indeed done dynamically depending on the return of every trajectory. This makes this method quite significant, especially when working with low-quality data.

* All the claims made by the authors are addressed through a comprehensive theoretical and empirical study. The paper is very well written; the notation is excellent, and all the details and formulas are clear.

* The experimental setup proposed is thorough. The paper analyzes how the modification of up to four state-of-the-art offline RL methods behaves across more than 25 different benchmarks.

* The theoretical analysis developed in Section 5 (and the corresponding annexes) is very robust and truly helps to understand what implementing HUBL entails.

**Weaknesses:**

* For someone not already familiar with offline RL, it can be challenging to follow the comprehensive theoretical analysis developed in this paper, especially in the appendices.

* This claim should be justified: "Despite their strengths, existing model-free offline RL methods also have a major weakness: they do not perform consistently." It would be fantastic if the authors could provide some evidences regarding this issue, as they do in section 3.2 (second to last paragrpah), but providing more details.

* In the experiments section I miss the learning curves where a reader can compare how the reward evolves with an without HUBL.

* I believe the manuscript needs a brief discussion on how the proposed model would be applied to problems with sparse rewards. For instance, in environments like Antmaze, has it been tested?

Minor comments:

-Section 4.1 HUBL introduce -> HUBL introduceS

**Questions:**

I've tried to detail most of the limitations and weaknesses of the proposed model in previous section, with some points that would need to be addressed in a rebuttal.

Overall, I see here a strong manuscript with some ideas that are adequate for an ICLR conference.

**Details Of Ethics Concerns:**

This work describes a model for learning agents from offline data. If the dataset is biased, the models will be biased too.

The work described in the manuscript was carried out in simulation and as such is unlikely to have produced unethical results, except the impact of large-scale training on CO2 output.

---

> ### Author Response · Authors · 2023-11-22
> **Thank you for your feedback. We have provided additional figures and experiments as you suggested.**
>
> Thank you very much for your constructive feedback and positive comment on our draft.
>
> > For someone not already familiar with offline RL, it can be challenging to follow the comprehensive theoretical analysis developed in this paper, especially in the appendices.
>
> Thank you for pointing out this presentation issue. We have uploaded an updated draft by adding more intuitions and explanations to the proof in Appendix A and B. Such modifications are highlighted in blue.
>
> > This claim should be justified: "Despite their strengths, existing model-free offline RL methods also have a major weakness: they do not perform consistently."
>
> We hope to justify our claim by two types of evidence:
> 1. Our experiments show that existing model-free offline RL methods show inconsistent performance over different tasks. See Appendix D.2 where provide the performance of each base offline RL algorithm on different tasks. Specifically, with the suggested hyperparameter tuning procedures, such offline RL methods especially underperform in some tasks while performing well in other tasks.
> 2. Such inconsistent performance can also be found in empirical results of other offline RL papers like Table 2 of [1]. In our experiments, we show that HUBL can mitigate this inconsistency issue and provide significant performance improvement in tasks where the base RL method underperforms. In the updated manuscript, we provide references for this statement.
>
> > In the experiments section I miss the learning curves where a reader can compare how the reward evolves with and without HUBL.
>
> We apologize for missing such a plot. As we focus on offline RL problems, the learned policy in each epoch does not need to interact with the environment. Therefore, when implementing the experiments, we only interacted with the environment at the end of the training to collect final performance results without documenting the performance in each epoch.
>
> Following your suggestion, **we redid the experiment of HUBL with TD3+BC on hopper-medium-v2**, and provided a figure of average normalized sum of rewards over time . Please see Figure 4 on page 23 of the updated submission.
>
> > I believe the manuscript needs a brief discussion on how the proposed model would be applied to problems with sparse rewards. For instance, in environments like Antmaze, has it been tested?
>
> Following your suggestion, we conducted additional experiments on applying HUBL with TD3+BC and Rank labeling to antmaze-medium-diverse-v0,  antmaze-medium-play-v0, antmaze-umaze-diverse-v0, and antmaze-umaze-v0. The results are provided below. HUBL is able to provide performance improvement to TD3+BC in the considered environments.
> |                           | Relative Performance Improvement of HUBL | Base |
> | ------------------------- | ---------------------------------------- | ---- |
> | antmaze-medium-diverse-v0 | 67%                                      | 30.0 |
> | antmaze-medium-play-v0    | 17%                                      | 60.0 |
> | antmaze-umaze-diverse-v0  | 61%                                      | 60.0 |
> | antmaze-umaze-v0          | 43%                                      | 70.0 |
>
> Theoretically, we did not pose any assumptions on whether the reward is sparse or not. Potentially, HUBL can be more helpful in sparse-reward environments if given a proper heuristic.  The reason is that even if the reward is sparse, the heuristic can be dense and guides decision-making. We defer a thorough analysis of HUBL in sparse-reward environments to future work. We will include such discussions in the final version of the submission.
>
> [1] Tarasov, Denis, et al. "CORL: Research-oriented deep offline reinforcement learning library." arXiv preprint arXiv:2210.07105 (2022).

---

### Official Review · Reviewer_8Sx2 · 2023-11-01

**Soundness:** 3 good
**Presentation:** 2 fair
**Contribution:** 3 good
**Rating:** 5
**Confidence:** 3

**Summary:**

The paper introduces the HUBL technique for offline RL algorithms, which partially replaces the bootstrapped values with heuristic ones estimated using Monte-Carlo returns.
Theoretical analysis has been made to understand the improvements brought by HUBL to the original offline RL algorithm, as well as its associated bias and regret.
Experimentally, results on the D4RL datasets and Meta-World benchmarks show that HUBL offers an average improvement of 9% over the current four SOTA offline RL algorithms.

**Strengths:**

* The proposed HUBL is a general technique that can be seen as a correction to the offline dataset itself, improving the performance of offline RL algorithms.
* Through theoretical analysis, the introduction of HUBL is discussed as an MDP reshaping, and the analysis of bias and regret is conducted.
* Extensive experiments empirically demonstrate that HUBL is indeed an effective enhancement technique.

**Weaknesses:**

* **Presentation**:
    * The presentation of the experimental results in the graphs lacks clarity. The absence of a horizontal baseline at 0 makes it unclear whether there's an improvement or decline. I believe a horizontal baseline at 0 should be added, and different colors could be considered to depict increases and decreases.
    * The experimental tables in the appendix have a similar problem. The best performances should be bolded for easier readability.
* **Limitations**:
As the authors discussed in the limitations section, offline datasets based on disconnected transition tuples are challenging to utilize with the HUBL trick unless heuristic values are computed during the construction of datasets.

**Questions:**

* Please improve the presentation of figures and tables in the paper as i mentioned above.
* I've noticed that on tasks where baseline offline RL algorithms already perform well, HUBL might decrease the performance. Is there a way to ensure that its enhancements are consistently non-negative?
* On D4RL, HUBL shows significant improvements over baseline algorithms for the hopper task several times. I wonder whether there exist any shared or general characteristics in situations where HUBL offers substantial advantages. Could this be discussed, or did I perhaps overlook any mention of this aspect?

---

> ### Author Response · Authors · 2023-11-22
> **Thank you for your feedback. The tables and plots have been updated as you suggested.**
>
> Thank you for your reply. We are grateful for your thorough reading and constructive comments, especially your suggestions on the presentation of experiment results.
>
> > Presentation
>
> We have uploaded an updated draft, where the plots and tables are modified as you suggested.
> For figures, a horizontal line at 0 is added to each plot, and bars are colored in red and blue, with blue for positive performance improvement and red for negative improvement.
> For tables, the best performing configuration in each row is highlighted in bold.
>
> > Limitations
>
> Although having trajectory data is a prerequisite of the current design of HURL, most real world decision-making data are actually collected as trajectories. Collecting independent transitions is rarely feasible. Thus, while we impose an assumption that might seem strong in the mathematical sense, it is actually weak in practice.
>
>
> >  Is there a way to ensure that its enhancements are consistently non-negative?
>
> Indeed, HUBL may occasionally hurt performance empirically on some datasets – this can happen in situations where offline RL algorithms already attain near-optimal results. Such performance decrease is more due to *finite sample error*, as the room for improvement on these datasets is small – for any offline learning technique. Nonetheless, in most medium-expert datasets where the base RL method already performs well, HUBL still provides positive performance improvement. Overall,  the main motivation behind HUBL is to consistently improve the performance of the base algorithm whenever the baseline algorithm performs suboptimally on its own (compared with other offline RL algorithms), and HUBL succeeds at this, sometimes with a relative improvement up to 50%.
>
> It is an interesting direction to design a method that can guarantee non-negative *expected* improvement. The tricky bit is that different offline RL methods have different characteristics like regret, approximation error, optimization error and so on. Therefore, it is hard to design one $\lambda$ that is rigorously guaranteed to improve expected performance for various offline RL methods without overtunning $\lambda$ for each problem.
>
>
> > On D4RL, HUBL shows significant improvements over baseline algorithms for the hopper task several times. I wonder whether there exist any shared or general characteristics in situations where HUBL offers substantial advantages.
>
> The empirical performance improvement of HUBL is most significant when the base offline RL method shows inconsistent performance and performs suboptimally compared to other offline RL algorithms. This can be seen from the results in Appendix D.2, where we provide the performance of the base RL algorithms in the experiments.

---

### Meta-Review · Area_Chair_XM9o · 2023-12-04

**Metareview:**

The paper proposes a novel and simple algorithm to improve offline RL techniques that perform bootstrapping.  The approach consists of relabling rewards and discount factors with heuristic values derived from Monte carlo returns.  The approach is analyzed theoretically and demonstrated by improving four state of the art offline RL techniques on 25 problems.  This represents a very nice contribution since the technique is simple to implement and yet quite effective as demonstrated by the experiments.

**Justification For Why Not Higher Score:**

While the approach clearly advances the state of the art both in terms of theory and practice, it is not ground breaking per se.

**Justification For Why Not Lower Score:**

The proposed approach clearly improves the state of the art and thanks to its simplicity, it will likely be broadly adopted by practitioners.

---

### Decision · Program_Chairs · 2024-01-16

Accept (spotlight)